# ZO-Offloading: Fine-Tuning LLMs with 100 Billion Parameters on a Single GPU

## Abstract

Fine-tuning pre-trained LLMs typically requires a vast amount of GPU memory. Standard first-order optimizers like SGD face a significant challenge due to the large memory overhead from back-propagation as the size of LLMs increases, which necessitates caching activations during the forward pass and gradients during the backward pass. In contrast, zeroth-order (ZO) methods can estimate gradients with only two forward passes and without the need for activation caching. Additionally, CPU resources can be aggregated and offloaded to extend the memory and computational capacity of a single GPU. To enable efficient fine-tuning of LLMs on a single GPU, we introduce ZO-Offloading, a framework that strategically utilizes both CPU and GPU resources for ZO. ZO-Offloading dynamically offloads model parameters to the CPU and retrieves them to the GPU as needed, ensuring continuous and efficient computation by reducing idle times and maximizing GPU utilization. Parameter updates are integrated with ZO's dual forward passes to minimize redundant data transfers, thereby improving the overall efficiency of the fine-tuning process. The ZO-Offloading framework also incorporates a novel low-bit precision technique for managing data transfers between the CPU and GPU in AMP mode, as well as asynchronous checkpointing for LLM fine-tuning. With ZO-Offloading, for the first time, it becomes possible to fine-tune extremely large models, such as the OPT-175B with over 175 billion parameters, on a single GPU with just 24GB of memory—a feat unattainable with conventional methods. Moreover, our framework operates without any additional time cost compared to standard ZO methodologies.

## 1 Introduction

As the scale of Large Language Models (LLMs) continues to grow, reaching parameter counts in the hundreds of billions like OPT-175B (Zhang et al., 2022) and Llama 3.1 405B (Dubey et al., 2024), managing GPU memory resources effectively becomes crucial. Efficient GPU memory management is crucial not only because it directly influences model performance and training speed, but also because GPU memory is both expensive and limited in quantity. However, this creates a significant challenge in handling ever-larger models within the physical constraints of current hardware technologies. CPU offloading has become a crucial technique for overcoming the challenge. It involves transferring computations and data from the GPU to the CPU, specifically targeting data or parameters that are less frequently accessed ("inactive"). Specifically, it leverages the typically larger and more cost-effective CPU memory (DDR SDRAM) compared to the more expensive and less abundant GPU memory (HBM). By offloading these inactive tensors of the neural network, CPU offloading effectively alleviates the memory and computational pressures on GPUs. While CPU offloading has been commonly applied in inference to manage memory-intensive tasks like KV cache offloading (Ge et al., 2023; Sheng et al., 2023) and Mixture of Experts (MoE) offloading (Eliseev & Mazur, 2023; Xue et al., 2024), its application in training, especially fine-tuning, remains less explored.

Recently, some works (Rajbhandari et al., 2020; Ren et al., 2021) have tried to introduce CPU offloading into LLM training. However, they are typically constrained by the capabilities of first-order optimizers such as SGD and Adaptive Moment Estimation (AdamW) (Loshchilov & Hutter, 2017), and limited GPU memory, restricting large-scale model scalability on single GPU systems. In detail, using first-order optimizers introduces two major inefficiencies in CPU offloading: **(1) Multiple**

**communication operations**: During the training of LLMs, parameters are used not only for computing the loss during the forward pass but also for gradient computation in the backward pass. This necessitates offloading the same data (parameter) twice—once for each pass (see Appendix Figure 5a for an illustration). Such redundancy not only doubles the communication volume between the CPU and GPU but also introduces significant latency and inefficiency due to repetitive data transfers. **(2) Huge data transfer volume per communication operation**: Furthermore, both parameters and activations (hidden states) are required in the backward pass to complete gradient computations. This means that parameters and activation values must be offloaded during each forward pass and re-uploaded to the GPU for the backward pass. The result is a significant increase in the volume of data transferred, which severely impacts training throughput and efficiency.

On the other hand, compared to first-order optimization methods, zeroth-order (ZO) methods offer a novel approach to fine-tuning LLMs (Zhang et al., 2024; Malladi et al., 2023; Gautam et al., 2024). These methods utilize dual forward passes to estimate parameter gradients and subsequently update parameters, as illustrated in Figure 5b. This approach eliminates the traditional reliance on backward passes, thereby streamlining the training process by significantly reducing the number of computational steps required.

Based on the above observations, we conjecture that ZO's architecture is particularly well-suited for CPU offloading strategies. Intuitively, by eliminating backward passes and the need to store activation values, it can significantly reduce GPU memory demands through efficient parameter offloading. However, despite these advantages, ZO training via CPU offloading introduces new challenges, particularly in the realm of CPU-to-GPU communication. Transferring parameters between the CPU and GPU, which is crucial for maintaining gradient computation and model updates, becomes a critical bottleneck due to inherent communication delays. Although ZO methods inherently extend computation times because of the dual forward passes, potentially allowing for better overlap between computation and communica-

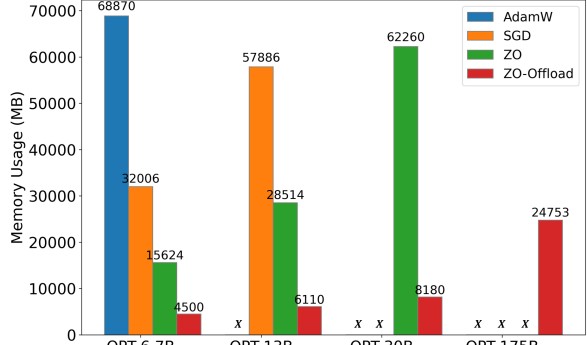

Figure 1: Single GPU memory usage comparison for training LLMs across different optimizers (AdamW, SGD, ZO, and ZO-Offloading) and model sizes (OPT-6.7B, OPT-13B, OPT-30B, OPT-175B). The 'X' indicates that training was not feasible due to excessive memory demand.

tion (Section 5.1), there remain significant inefficiencies. The necessity to upload parameters to the GPU for upcoming computations introduces a large volume of communications. Additionally, when employing Automatic Mixed Precision (AMP) (Micikevicius et al., 2017) training, which accelerates computation using NVIDIA's Tensor Cores [1], the discrepancy between the rapid computation and slower communication phases is further magnified as AMP only accelerates the computation but does not accelerate the communication. This is because, although AMP computes using a faster bit format, the underlying storage format retains its original bit width. Consequently, the volume of data communicated remains unchanged.

To tackle the inefficiencies highlighted, we introduce ZO-Offloading, a novel framework specifically designed for ZO fine-tuning in LLMs with CPU offloading. This framework utilizes the unique dual forward pass architecture of ZO methods to optimize interactions between CPU and GPU, significantly enhancing both computational and communication efficiency. By building a high-performance dynamic scheduler, ZO-Offloading achieves substantial overlaps in communication and computation. Our strategy further integrates AMP training, which not only improves computation throughput but also incorporates low-bit weight compression during both parameter uploads and offloads, further reducing the data transfer volume necessary for AMP training. To enhance practical usability and efficiency, we also propose asynchronous checkpointing for ZO LLM training. These innovations make it feasible to fine-tune extremely large models, such as the OPT-175B (Zhang et al., 2022) with **over 175 billion parameters, on a single GPU equipped with just 24GB of memory**—a capability previously unattainable with conventional methods (Figure 1). Ad-

---

[1]https://www.nvidia.com/en-us/data-center/tensor-cores/

ditionally, our efficient framework operates without any extra time cost and decreases in accuracy compared to standard ZO methodologies. Our contributions can be summarized as follows:

- **Innovative use of CPU-offloading for ZO methods**: We pioneer the application of CPU offloading in the context of ZO optimization methods to dramatically reduce GPU memory requirements. This method allows for the efficient handling of model parameters by dynamically transferring inactive data between the CPU and GPU, significantly extending the capacity to train large models like OPT-175B on a single GPU.
- **Low memory but high-throughput framework**: We introduce a series of optimized features that substantially reduce GPU memory use while maintaining high throughput. Our dynamic scheduler improves GPU utilization by optimizing computation and communication overlaps. Reusable memory blocks minimize overhead and stabilize memory use, while efficient parameter updating synchronizes updates with dual forward passes to reduce data transfers. Extended AMP support and asynchronous checkpointing boost computational speed and reduce training interruptions, ensuring efficient training on constrained hardware with minimal memory footprint.
- **Empirical Validation and Experimentation**: Our experiments demonstrate that ZO-Offloading can efficiently fine-tune the OPT-175B model, with over 175 billion parameters, on a single 24GB GPU—previously impossible with traditional methods. Crucially, this is achieved with no additional time cost and decreases in accuracy, showcasing the framework's effectiveness and efficiency for large-scale model training.

## 2 RELATED WORK

**Zeroth-Order (ZO) Optimization.** ZO optimization offers a gradient-free alternative to first-order (FO) optimization by approximating gradients through function value-based estimates. These estimates theoretically require only two forward passes but are believed to be prohibitively slow for optimizing large models. Despite this limitation, ZO methods have been utilized in deep learning to generate adversarial examples or adjust input embeddings (Sun et al., 2022a;b), though they have not been widely adopted for direct optimization of large-scale models (Liu et al., 2020). Several acceleration techniques have been proposed to address the scaling challenges of ZO optimization and some of them have been used for LLM fine-tuning. These include using historical data to improve gradient estimators (Cheng et al., 2021), exploiting gradient structures (Singhal et al., 2023) or sparsity to reduce the dependence of ZO methods on the size of the problem (Chen et al., 2024; Cai et al., 2022; 2021), and reusing intermediate features (Chen et al., 2024) and random perturbation vectors (Malladi et al., 2023) during the optimization process. These advancements suggest that ZO optimization could increasingly be applied to more complex and large-scale ML problems. While previous ZO optimization efforts have primarily targeted algorithmic improvements for GPU memory efficiency, our approach extends these optimizations to the system level, enabling more robust memory management and enhanced performance for large-scale machine learning applications.

**CPU Offloading for LLMs.** With recent advancements in LLMs, several approaches have emerged to offload data to CPU memory, mitigating GPU memory limitations. One such method is vLLM (Kwon et al., 2023), which utilizes PagedAttention to dynamically manage the key-value (KV) cache at a granular block level. Portions of the KV cache can be temporarily swapped out of GPU memory to accommodate new requests. Llama.cpp (Gerganov, 2023) addresses oversized LLMs by using static layer partitioning. It stores certain contiguous layers in CPU memory while keeping others in GPU memory. During computation, the CPU handles the layers in its memory, followed by the GPU computing its assigned layers. FlexGen (Sheng et al., 2023), a GPU-centric inter-layer pipeline method, seeks to improve throughput by pinning some model weights in GPU memory for each layer. During computation, it overlaps GPU processing of the current layer with data loading for the next. DeepSpeed (Rajbhandari et al., 2020) introduces a technique to offload the first-order optimizer state to the CPU, significantly reducing GPU memory requirements during training. Zero-offload (Ren et al., 2021) extends the DeepSpeed approach by not only offloading data to the CPU but also engaging the CPU in computational tasks. Despite these advancements, the predominant focus of previous research has been on optimizing LLM inference or first-order optimization through strategic CPU-GPU data transfers. Our work, in contrast, introduces a novel approach by implementing CPU offloading specifically for zeroth-order optimization and fine-tuning of LLMs.

## 3 PRELIMINARIES ON ZO AND ZO-SGD

ZO optimization offers a gradient-free alternative to first-order (FO) optimization by approximating gradients through function value-based estimates. There are different ZO optimizers for estimating the gradient. To better illustrate our framework, in this paper, we focus on the randomized gradient estimator (RGE) proposed by (Nesterov & Spokoiny, 2017), which approximates the FO gradient using finite differences of function values along randomly chosen direction vectors and has been used widely in the ZO optimization literature. Our idea can be applied to other ZO optimizers.

Given a scalar-valued function $f(\cdot)$ and a model $x$ with parameters in $d$ dimensions, the RGE employed by (Malladi et al., 2023), referred to as $\hat{\nabla} f(x)$, is to approximate $\nabla f(x)$ and is expressed using central difference:

$$\hat{\nabla} f(x) = gz \in \mathbb{R}^d, \tag{1}$$

$$g = \frac{f(x + \epsilon z) - f(x - \epsilon z)}{2\epsilon} \in \mathbb{R}^1, \tag{2}$$

where $z$ is a random direction vector drawn from the standard Gaussian distribution $\mathcal{N}(0, \mathrm{I})$, and $\epsilon > 0$ is a small perturbation step size, also known as the smoothing parameter. $g$ represents the projected gradient computed using the model's dual-forward passes. Notably, $g \in \mathbb{R}^1$ is just a scalar value and requires minimal memory space. The rationale behind RGE stems from the concept of the directional derivative (Duchi et al., 2015). As $\epsilon$ approaches 0, the directional derivative provides us an unbiased gradient estimator of $\nabla f(x)$. Thus, the RGE $\hat{\nabla} f(x)$ can be interpreted as an approximation of the FO gradient $\nabla f(x)$ using the directional derivative (Zhang et al., 2024). Zeroth-order stochastic gradient descent (ZO-SGD) follows a similar algorithmic framework to its first-order counterpart, SGD, but replaces the gradient with an estimated gradient via zeroth order (function value) information for the descent direction.

Fine-tuning pre-trained LLMs typically demands substantial GPU memory. Previous first-order methods encounter major challenges as LLM sizes grow, primarily due to the significant memory overhead required for backpropagation, which involves storing activations during the forward pass and gradients during the backward pass. In contrast, ZO can estimate gradients with only forward passes, eliminating the need for activation caching. (Malladi et al., 2023) utilized the classical ZO algorithm (based on RGE), named MeZO, to fine-tune pre-trained LLMs with up to 30 billion parameters on a single GPU. They capitalized on the memory-efficient nature of ZO optimization, which eliminates the need for backpropagation and reduces memory costs. Since CPU resources can be combined and offloaded, the memory and computational capacity of the GPU can be expanded. To facilitate efficient fine-tuning of LLMs on a single GPU, we introduce ZO-Offloading, a framework that strategically leverages both CPU and GPU resources for ZO.

## 4 ZO-OFFLOADING FRAMEWORK

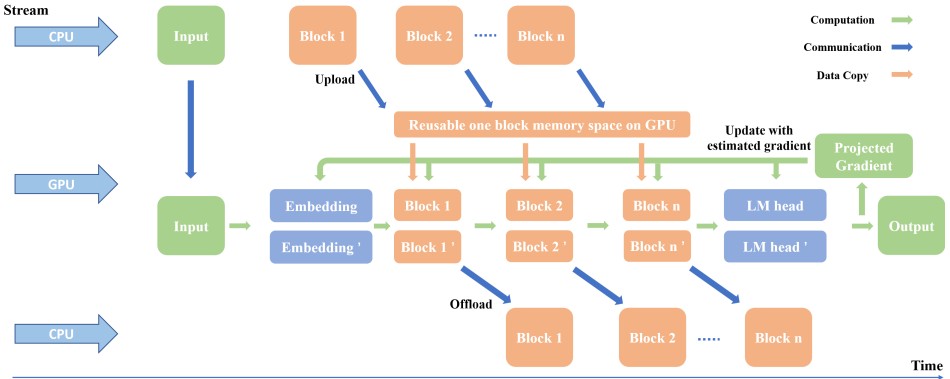

Figure 2: Workflow of the ZO-Offloading framework (non-AMP) for fine-tuning LLMs.

In this section, we first provide an overview and a brief introduction to our ZO-Offloading framework. To better illustrate our idea, we first describe the computation workflow of the original ZO optimization procedure for LLM fine-tuning. Initially, input data is loaded from the disk into the CPU

and subsequently transferred to the GPU. Within the GPU, each module—including the embedding layer, transformer blocks, and the language model (LM) head—executes dual forward computations to estimate the projected gradient and update parameters. From the system perspective, traditional deep learning frameworks like PyTorch (Paszke et al., 2019) typically manage both communication (via interconnections, e.g., PCIe) and computation tasks with a single CUDA stream[2], leading to significant inefficiencies. Specifically, for ZO optimization, the $i$-th transformer block is uploaded from the CPU to the GPU (the GPU is designated for computation-intensive tasks using its CUDA and Tensor Cores, and the CPU memory is used for parameter storage), undergoes dual forward computation, and then is offloaded back to the CPU. The $i+1$-th block must wait for the offloading of the $i$-th block to finish before its uploading, leading to idle CUDA and Tensor Cores during communication while the interconnection remains idle during computation. See Figure 6 in Appendix for an illustration.

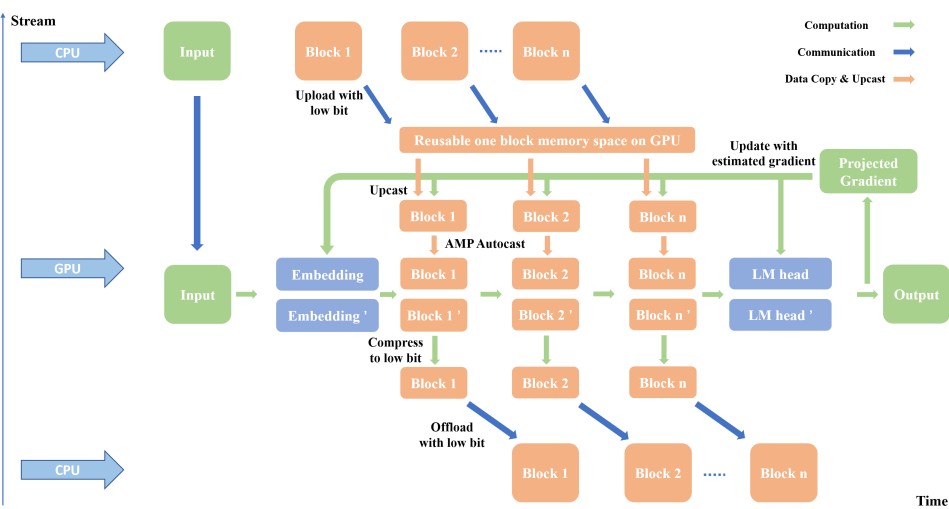

Figure 3: Workflow of the ZO-Offloading framework (AMP mode) for fine-tuning LLMs.

Central to our ZO-Offloading framework is the strategic utilization of CPU and GPU resources (Section 5.1). This approach involves dynamically offloading model parameters to the CPU and uploading them back to the GPU as needed for computation. Specifically, for the transformer model structure, each transformer block is individually uploaded for processing and subsequently offloaded post-computation, thus balancing communication and computation across blocks. As illustrated in Figure 2, while the $i$-th transformer block is being computed, the $i+1$-th block is pre-uploaded, and the $i-1$-th block is offloaded simultaneously. This strategic overlapping ensures continuous and efficient computation, reducing idle times and maximizing GPU utilization. In the uploading phase of ZO-Offloading, transformer blocks are transferred into a reusable memory space on the GPU, eliminating the extra time typically required for CUDA memory allocation (Section 5.2). Moreover, parameter updates are ingeniously fused with the dual forward passes to minimize redundant data transfers, thereby enhancing the overall efficiency of the model training process (Section 5.3).

Our ZO-Offloading framework further integrates a novel low-bit precision technique that efficiently manages data transfers between the CPU and GPU in the AMP mode (see Figure 3 for an illustration). This technique is aligned with AMP protocols by ensuring that high-bit precision is maintained for parameter updates, while low-bit precision data is used for computation on the GPU (Section 5.4). This dual-precision approach significantly reduces the communication overhead, optimizing memory usage without compromising computational accuracy. The adoption of low-bit compression during both the upload and offload phases further minimizes the data transfer volume, streamlining the training process and allowing for the efficient handling of large-scale models on constrained hardware setups.

In the following section, we will provide challenges and details of our framework.

---

[2]https://developer.download.nvidia.com/CUDA/training/StreamsAndConcurrencyWebinar.pdf

## 5 DESIGN AND IMPLEMENTATION DETAILS

---

**Algorithm 1** ZO-Offloading Dynamic Scheduler

---

**Require:** Transformer blocks $\{W_i\}_{i=1}^N$ with number of transformer blocks $N$, embedding parameters $Embedding$, and LM head $LMhead$.

1: Initialize a dynamic scheduler $S\{\cdot\}$ to control dual forward computation $C(\cdot)$, uploading $U(\cdot)$, and offloading $O(\cdot)$ operations.
2: Asynchronously launch $S\{U(W_1), C(Embedding)\}$.
3: **for** $i = 1$ to $N - 1$ **do**
4:     Synchronously wait until $U(W_i)$ finished.
5:     **if** $i = 1$ **then**
6:         Asynchronously launch $S\{U(W_{i+1}), C(W_i)\}$.
7:     **else**
8:         Synchronously wait until $C(W_{i-1})$ finished.
9:         Asynchronously launch $S\{U(W_{i+1}), C(W_i), O(W_{i-1})\}$.
10:    **end if**
11: **end for**
12: Synchronously wait until $U(W_N)$ and $C(W_{N-1})$ finished.
13: Asynchronously launch $S\{C(W_N), O(W_{N-1})\}$.
14: Synchronously wait until $C(W_N)$ finished.
15: Asynchronously launch $S\{C(LMhead), O(W_N)\}$.

---

### 5.1 DYNAMIC SCHEDULER DESIGN FOR EFFICIENT OVERLAP

To overlap the data loading and computation process, we propose a dynamic scheduler, utilizing the asynchronous execution on different CUDA streams. Specifically, our scheduler includes three CUDA streams (Figure 2), which are utilized to control the $i$-th transformer block's computation, the $i + 1$-th block's uploading, and the $i - 1$-th block's offloading can occur concurrently. This design minimizes data transfer conflicts and maximizes GPU utilization by keeping computational and communication channels active.

However, designing this dynamic scheduler presents challenges when communication tasks outlast computation tasks, leading to potential errors. For example, if the upload of the $i$-th block is incomplete when its computation begins, this can lead to errors, as the GPU computes with an incomplete set of parameters. Similarly, if the computation of the $i$-th block is still ongoing when its offloading begins, it can also result in errors because the computation is disrupted by the removal of necessary data. To address this, our scheduler implements a locking mechanism for each block's computation task, ensuring it only starts once its corresponding upload is confirmed complete. While this solution mitigates the issue of incomplete parameters, it can still potentially create bottlenecks if communication tasks consistently outlast computation tasks. Surprisingly, our evaluations show that with ZO's unique dual forward passes, which extend computation times, communication delays are no longer the primary bottleneck in most scenarios.

Moreover, special attention needs to be given to the embedding parameters and the LM head, as they represent the beginning and end of the model, respectively. By consistently maintaining both the embedding and LM head on the GPU, we circumvent the overhead linked to frequent transfers. For the embedding layer, simultaneous uploading of input data and embedding parameters could compete for interconnection bandwidth. Moreover, keeping the embedding layer on the GPU enables the pre-uploading of the first transformer block, effectively overlapping with the computations of the embedding layer. Meanwhile, continuously keeping the LM head on the GPU removes delays associated with its offloading—since no subsequent block computations overlap with this offloading—and facilitates weight sharing with the embedding layer, as noted in some conditions (Radford et al., 2019), thus consolidating related computations and enhancing operational efficiency. The detailed scheduler design to apply ZO-Offloading on LLMs is shown in Algorithm 1.

### 5.2 EFFICIENT MEMORY MANAGEMENT VIA REUSABLE ONE BLOCK SPACE ON GPU

We can further optimize memory management by initially pre-allocating a reusable transformer block of memory on the GPU. This strategy is implemented to circumvent the substantial time overhead associated with repeated CUDA memory allocations (malloc) and frees, which are typi-

cally required each time when data is transferred between the CPU and the GPU. By establishing a dedicated memory space initially and reusing it for each transformer block, we avoid the need for multiple malloc and free operations overhead the training process.

This reusable memory space is dynamically assigned to accommodate the parameters of each transformer block sequentially. Once a block's computation is complete and its data is offloaded back to the CPU, the same GPU memory space is immediately prepared to receive the next block's parameters from the CPU. This approach not only expedites the data transfer process but also stabilizes the GPU's memory usage, preventing fluctuations that could otherwise impact computational efficiency and performance.

### 5.3 EFFICIENT PARAMETER UPDATE STRATEGY

In the ZO-Offloading framework, the parameter update strategy is meticulously designed to precede the dual forward computations of each transformer block. Traditionally, each transformer block is subjected to two distinct data transfer phases (Figure 7a): one for the dual forward computations and another for applying gradient updates. This requirement stems from the fact that the (approximated) gradients are obtained only after completing the dual forward computations for the entire model. Consequently, parameters must be uploaded for the computation phase, offloaded upon completion, and then re-uploaded and offloaded again for the gradient update phase. This iterative process effectively doubles the communication load and extends the duration of training.

By implementing preemptive parameter updates, the framework significantly curtails the number of data transfers required per iteration (Figure 7b). With this strategy, once blocks are updated with the last iteration's gradients, only a single upload and offload cycle is necessary for each block. This adjustment not only halves the usage of interconnection bandwidth but also enhances the efficiency of the training process, thereby streamlining operations and reducing overhead.

### 5.4 ZO-OFFLOADING IN AMP MODE

Figure 3 illustrates the workflow of the ZO-Offloading framework under AMP mode, which employs reduced precision formats to accelerate the training of LLMs. AMP leverages formats such as Tensor Float Point 32 (TF32), which provides higher computational throughput compared to Float Point 32 (FP32). This acceleration is critical for enhancing training efficiency but introduces challenges in maintaining effective computation-communication overlap, as the data transfer still utilizes the FP32 format.

To address this, the ZO-Offloading framework incorporates a compression mechanism where parameters are compressed to low-bit formats during offloading from GPU to CPU. This compression significantly reduces the data volume, enabling quicker transfers and mitigating bandwidth limitations. The current compression settings include bfloat16 and float16, which reduce the data size by 50%, and more aggressive reductions like float8, which compress to 25% of the original size.

Upon uploading these compressed parameters back to the GPU, they are decompressed and restored to FP32 for high-precision parameter updates. Subsequent computations, particularly the dual forward passes, are then performed using the TF32 format to exploit the computational speed.

### 5.5 EXTENSION: ASYNCHRONOUS CHECKPOINTING

Checkpointing (Rojas et al., 2020) is an indispensable technique in the training of LLMs, acting as a critical safeguard against data loss and enabling the resumption of training from specified states. This process involves periodically saving the state of the model to disk, which becomes increasingly frequent as the model size increases. This is essential for preserving significant progress in model training but introduces substantial computational and communication challenges. Traditionally, checkpointing a large-scale LLM interrupts ongoing computations as the model is transferred from the GPU to the CPU and subsequently saved to disk. This can be exceedingly time-consuming; for instance, employing `torch.save()` to checkpoint an 11-billion-

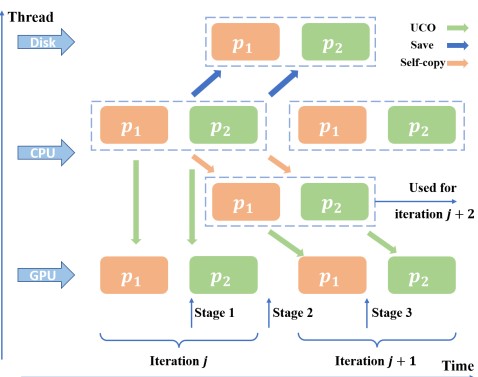

Figure 4: Asynchronous checkpointing.

parameter model can take up to 30 minutes.[3] The
delays are primarily due to the extensive data involved and the limited bandwidth available for data transfer.

**Asynchronous checkpointing.** In the ZO-Offloading framework, we exploit the fact that most parameters are already stored on the CPU, eliminating the need for a GPU-to-CPU offload during checkpointing. However, the time required to transfer data from the CPU to the disk remains significant, often exceeding the time it would take to offload data from the GPU to the CPU.

To address these delays, we have developed a strategy for asynchronous checkpointing (Figure 4) that allows training to continue uninterrupted. Specifically, the model parameters are conceptually divided into two equal partitions: $p_1$ and $p_2$. $p_1$ contains the first half of the whole transformer blocks and the embedding module, while $p_2$ includes the second half and the LM-head, maintaining the integrity and order of the parameters. The dashed boxes in the figure represent the complete model parameters. Checkpointing is initiated when $p_1$ has completed its cycle of uploading, computation, and offloading (UCO), but before $p_2$ begins its cycle. This timing ensures that each part of the model can be handled independently in terms of data saving.

The asynchronous checkpointing process is structured in three stages: **Stage 1**: Upon initiating checkpointing in $j$-th training iteration, the scheduler launches multiple threads to handle three tasks concurrently: saving $p_1$ from CPU to disk, creating a self-copy of $p_1$, and managing the UCO cycle of $p_2$. **Stage 2**: As the model progresses to the $j + 1$ training iteration, the scheduler waits for the completion of $p_1$'s self-copy to ensure data integrity, then asynchronously initiates the UCO cycle for $p_1$, while simultaneously saving $p_2$ to disk and creating a self-copy of $p_2$. **Stage 3**: The scheduler waits for $p_2$'s self-copy to complete before launching its UCO cycle.

The inclusion of self-copying stages is designed to safeguard against potential delays in saving to disk. Self-copying is not only faster than transferring data from the CPU to the disk but also quicker than the UCO cycles of $p_1$ or $p_2$. By the end of the $j + 1$-th iteration, the entire model is copied on the CPU, ready for immediate use in the $j + 2$-th iteration's UCO process without the need for re-uploading from the disk. However, it is notable that although it increases throughput, this asynchronous checkpointing method introduces a trade-off by increasing CPU memory usage.

## 6 EXPERIMENT

The experimental evaluation of our framework was conducted using the PyTorch deep learning library, integrated with NVIDIA CUDA streams to optimize parallel computation tasks. We selected the Open Pre-trained Transformer (OPT) (Zhang et al., 2022) model family as the subject of our experiments due to its open-source availability, widespread adoption in the research community, and diverse range of model sizes, ranging from 125 million to 175 billion parameters, which allows for a comprehensive assessment of our framework's performance across varying scales of model complexity. In our evaluation, MeZO serves as the baseline method, as it is the most memory-throughput efficient ZO method currently. Our framework builds upon MeZO, reducing GPU memory usage while maintaining throughput and precision. All performance evaluation experiments are done with dataset SST-2 (Socher et al. (2013)). Additional experimental settings, the evaluation of asynchronous checkpointing, and more extra experiments are included in Appendix C and D.

### 6.1 MAIN RESULTS

The performance results of our experiments are presented in Table 1, where we compare the GPU memory usage and throughput of the MeZO and ZO-Offloading frameworks, employing both FP32 and FP16 data formats. The results demonstrate a consistent advantage of ZO-Offloading in terms of GPU memory utilization across all model sizes, highlighting significant efficiency improvements, especially in large-scale models like **OPT-175B**. This efficiency is attributed to ZO-Offloading's design, which strategically utilizes GPU memory to temporarily store only a limited number of transformer blocks for computation rather than the entire model. Notably, the memory savings become more pronounced as the model size increases. For smaller models, the GPU memory savings

---

[3]https://pytorch.org/blog/reducing-checkpointing-times/

Table 1: **Main results of ZO-Offloading performance for various model configurations and both FP32 and FP16 modes.** Instances of '-' in the table indicate scenarios where the corresponding method failed to execute due to memory constraints. The values in parentheses (x) represent the ratio of each measurement compared to the baseline MeZO (first column) configuration.

| Model | GPU Memory Usage (MB) ↓ | | | | Throughput (tokens/sec) ↑ | | | |
|---|---|---|---|---|---|---|---|---|
| | MeZO(32) | ZO-Offload(32) | MeZO(16) | ZO-Offload(16) | MeZO(32) | ZO-Offload(32) | MeZO(16) | ZO-Offload(16) |
| OPT-125M | 3091 | 2941(x0.95) | 1801(x0.58) | **1661(x0.54)** | 14889 | 13074(x0.89) | **31058(x2.09)** | **31058(x2.09)** |
| OPT-350M | 4219 | 3393(x0.81) | 2389(x0.57) | **1643(x0.39)** | 5274 | 5099(x0.97) | **13508(x2.56)** | 12284(x2.32) |
| OPT-1.3B | 9117 | 4413(x0.48) | 4887(x0.54) | **2651(x0.29)** | 1954 | 1954(x1.00) | **6788(x3.47)** | **6788(x3.47)** |
| OPT-2.7B | 15277 | 5261(x0.34) | 7933(x0.52) | **3111(x0.20)** | 1087 | 1087(x1.00) | **4227(x3.89)** | **4227(x3.89)** |
| OPT-6.7B | 32083 | 8329(x0.26) | 16311(x0.51) | **4539(x0.14)** | 499 | 499(x1.00) | **2455(x4.92)** | **2455(x4.92)** |
| OPT-13B | 58251 | 12113(x0.21) | 29411(x0.50) | **6445(x0.11)** | 270 | 270(x1.00) | **1406(x5.21)** | 1340(x4.96) |
| OPT-30B | - | 18879 | 63953 | **10369** | - | 122 | **651** | 597 |
| OPT-66B | - | 29937 | - | **14143** | - | 40 | - | **273** |
| OPT-175B | - | 49203 | - | **24667** | - | 14 | - | **37** |

are less pronounced due to the significant proportion of memory allocated for input data, which diminishes the relative impact of the memory optimization.

In terms of throughput, ZO-Offloading maintains a performance comparable to MeZO in most tested scenarios without any additional time overhead. The instances where ZO-Offloading exhibits a decrease in throughput, such as with the OPT-125M model in FP32 format, can be primarily attributed to the dynamics of computation and communication. In these cases, the computation of each transformer block's dual forward passes completes quicker than their corresponding communication tasks, leading to idle times as the dynamic scheduler (discussed in Section 5.1) synchronizes and waits for these communication tasks to conclude. It is important to note that our results do not show a consistent pattern where either smaller or larger models benefit more significantly from the computation-communication overlap, indicating that the effectiveness of this overlap does not linearly correlate with model size.

Additionally, our method should maintain accuracy compared to MeZO, as we did not alter the underlying computation of ZO optimization. We conducted accuracy verification experiments to confirm this. The results of these experiments are detailed in Table 4 in the Appendix. These tests affirm that our ZO-Offloading method preserves model accuracy across different model sizes and data formats, reinforcing the robustness of our approach.

## 6.2 ABLATION STUDY OF SCHEDULER, REUSABLE MEMORY, AND EFFICIENT UPDATING

Table 2: **Throughput (token/sec) results to validate proposed features.**

| Model | MeZO | ZO-Offloading (no scheduler overlap) | ZO-Offloading (no reusable memory) | ZO-Offloading (no efficient update) | ZO-Offloading |
|---|---|---|---|---|---|
| OPT-1.3B | 1954 | 1109 (x0.57) | 735 (x0.38) | 1567 (x0.80) | 1954 (x1.00) |
| OPT-2.7B | 1087 | 573 (x0.52) | 422 (x0.39) | 849 (x0.78) | 1087 (x1.00) |
| OPT-6.7B | 499 | 225 (x0.45) | 184 (x0.37) | 373 (x0.74) | 499 (x1.00) |

In order to discern the individual contributions of key features within the ZO-Offloading framework to its overall performance, an ablation study was conducted focusing on three critical components: the dynamic scheduler (Sec. 5.1), reusable memory (Sec. 5.2), and efficient parameter updating (Sec. 5.3). This study mainly focused on throughput because the primary objective of the three features under investigation was to enhance throughput without impacting ZO-Offloading's inherent capability to reduce GPU memory usage. The main results, as presented earlier, clearly demonstrated that ZO-Offloading effectively decreases GPU memory consumption. Therefore, an ablation study on memory usage was deemed unnecessary, as the CPU-offloading mechanism inherently manages to reduce memory demands without the need for additional features aimed specifically at memory reduction. Given the tightly integrated nature of our system, traditional ablation methodologies that add one feature at a time to a baseline are impractical. Instead, we adopted a reverse ablation approach where each feature was individually disabled. This allowed us to observe the decrement in throughput relative to the fully operational framework, thereby highlighting the significance of each component. We mainly use OPT-1.3B, OPT-2.7B, and OPT-6.7B in the ablation study. The ablation study of more models is included in the Appendix (Table 5).

The results, presented in Table 2, provide a clear illustration of how the absence of each feature impacts the system's throughput: (1) **Horizontal Comparison.** Across all models, the removal of reusable memory results in the most substantial decrease in throughput, followed by the dynamic scheduler, and finally, the efficient parameter updating. This order of impact suggests that while all three features are pivotal, the overhead introduced by CUDA malloc operations, which are eliminated by reusable memory, significantly outweighs the communication delays between the CPU and GPU, managed by the dynamic scheduler and efficient parameter updating. For instance, when reusable memory is not employed, the throughput drops to 37% of the fully optimized framework for the OPT-6.7B model, highlighting its critical role in enhancing performance. (2) **Vertical Comparison.** As the model size increases, the relative importance of the dynamic scheduler and efficient parameter updating grows more pronounced. This trend is observable from the throughput: for larger models like OPT-6.7B, the reduction in throughput when the scheduler and efficient update features are disabled is relatively larger than in small models. This indicates that as models become larger, the complexities and overheads associated with managing and optimizing communications between CPU and GPU become more critical to maintaining performance. Conversely, the impact of reusable memory remains relatively constant across different model sizes, reinforcing the idea that while CUDA malloc operations are significant, their relative burden does not scale in the same way as communication overheads.

### 6.3 EVALUATION OF AMP MODE

Table 3: **Throughput (token/sec) results to validate AMP Mode.** AMP auto-cast with FP16 (top) and BF16 (below).

| Model | ZO-Offload (non-compress) | ZO-Offload (FP16) | ZO-Offload (BF16) | ZO-Offload (FP8) |
|---|---|---|---|---|
| OPT-1.3B | 4827 | 4770 (x0.988) | 4760 (x0.986) | 4802 (x0.995) |
| OPT-2.7B | 2811 | 2974 (x1.058) | 2974 (x1.058) | 2997 (x1.066) |
| OPT-6.7B | 1271 | 1641 (x1.291) | 1641 (x1.291) | 1662 (x1.308) |
| OPT-1.3B | 4565 | 4430 (x0.970) | 4430 (x0.970) | 4463 (x0.978) |
| OPT-2.7B | 2778 | 2816 (x1.014) | 2816 (x1.014) | 2818 (x1.014) |
| OPT-6.7B | 1273 | 1594 (x1.252) | 1594 (x1.252) | 1612 (x1.266) |

The efficiency of the AMP mode is shown in Table 3, where we evaluate the throughput using two AMP auto-cast computational data formats: FP16 and BF16. Additionally, we investigate the impact of various compression formats (FP16, BF16, and FP8) on communication and computation performance as detailed in Section 5.4.

Across all models tested, a clear trend emerges: lower-bit compression formats consistently yield higher throughput. Notably, there is no significant difference in throughput between the 16-bit formats, FP16 and BF16, suggesting that the compression efficiency rather than the specific format type is the crucial factor in enhancing communication speed.

In most scenarios (specifically for the OPT-2.7B and OPT-6.7B models), employing low-bit compression results in superior throughput, underscoring the benefits of reducing data transfer volumes. However, exceptions are observed, such as with the OPT-1.3B model, where non-compressed data slightly outperforms the compressed formats. This outcome is attributed to the system being computation-bound rather than communication-bound. In such contexts, the additional computational demands imposed by the compression process do not sufficiently offset the benefits of reduced data transfer times, thereby introducing an overhead that detracts from the overall system efficiency.

## 7 CONCLUSION

In this paper, we presented ZO-Offloading, an efficient framework that enables the training of extremely large language models, such as the OPT-175B, on a single 24GB GPU—a capability previously unattainable with traditional methods. By effectively integrating CPU offloading, high-performance dynamic scheduler, efficient memory management, efficient parameter updating, AMP support, and asynchronous checkpointing, our framework reduces GPU memory demands while maintaining high throughput without additional time costs. These innovations not only lower the bar for teams with limited hardware resources and advance the democratization of large models, but also open new avenues for advancing AI technology more efficiently. Moving forward, we plan to further enhance ZO-Offloading, exploring synergies with emerging hardware and optimization techniques to keep pace with the evolving demands of AI model training.

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
