## A  DISCUSSION

This section discusses several pertinent questions that might arise from the ZO-Offloading framework, providing deeper insights into the design decisions and operational nuances of the system.

- **CPU offloading does not excessively consume CPU memory.** Typically, in traditional PyTorch training, CPU memory is inevitably consumed to accommodate the model's parameters, as both the model's initialization and its subsequent storage necessitate CPU memory allocation.

- **The disk offloading strategy has been abandoned.** Although our asynchronous check-pointing could inspire the disk offloading strategy to extend CPU offloading, prior experimentation with disk offloading revealed that the latency involved in disk-CPU-GPU communication significantly hampers performance—occasionally, the time taken for a single block's communication exceeds the total computation time of the model on the GPU. Our goal is to maximize throughput without compromising it through offloading. Consequently, we have abandoned the disk offloading strategy.

- **The multi-GPU strategy is not adopted.** The primary aim of this paper is to reduce reliance on GPU memory by leveraging increased CPU memory instead. We believe that the current system architecture adequately supports most model sizes (up to 175 billion parameters) without the need for expanding to multiple expensive GPUs.

## B  ADDITIONAL DETAILS ON MOTIVATIONS AND PREVIOUS APPROACHES

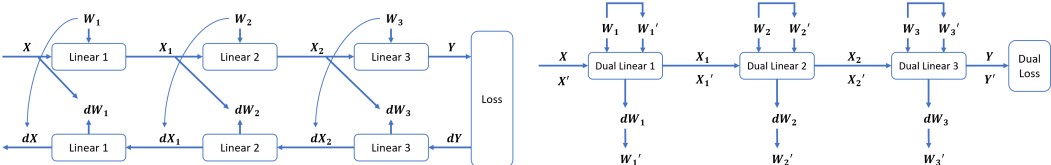

(a) Model using first-order optimizer with forward-backward passes workflow

(b) Model using zero-order optimizer with only forward passes workflow

Figure 5: **Motivation**. Comparison of model workflows using first-order and zeroth-order optimizers. (a) depicts a traditional first-order optimizer workflow with forward and backward passes, while (b) shows a zeroth-order optimizer workflow utilizing only forward passes.

**Why ZO is Suitable for CPU Offloading** Figure 5 illustrates the distinct operational differences between first-order and zeroth-order optimization methods applied to model training. Figure 5(a) demonstrates a traditional first-order optimizer setup, where the model employs a forward-backward pass sequence to update weights. Here, the input $X$ progresses through several linear transformations (Linear 1, 2, 3), generating intermediate activations $(X_1, X_2)$ and the final output $Y$, which is used to compute the loss. Subsequent backward passes calculate gradients $(dW_1, dW_2, dW_3)$ for each weight and derivatives for each activation $(dX, dX_1, dX_2)$, necessary for parameter updates through gradient descent. In this setup, each parameter $W$ is offloaded from the GPU to the CPU after the forward computation but requires reloading during backpropagation, resulting in dual transfers for each parameter in the computation process. Additionally, activations consume significant GPU memory.

In contrast, Figure 5(b) presents the zeroth-order optimizer's workflow, which simplifies the training process by eliminating the backward passes. This setup involves dual forward passes through slightly perturbed versions of the model weights $(W_1', W_2', W_3')$ at each layer (Dual Linear 1, 2, 3). The resulting outputs from each layer $(X', X_1', X_2')$ and the final output $Y'$ are used to compute a dual loss. This dual loss approximates the gradient required for updating the original weights, relying solely on forward computations. This approach not only reduces computational overhead and memory demands by obviating the need to store activations but also enhances efficiency by requiring only a single transmission of each parameter $W$ during the entire computational flow—-from the GPU to the CPU after its final usage in the dual forward passes—-thereby eliminating the need

for subsequent reloading during backpropagation. The ZO method's reliance on forward-only computations and efficient CPU offloading significantly benefits the training of large models on limited hardware setups.

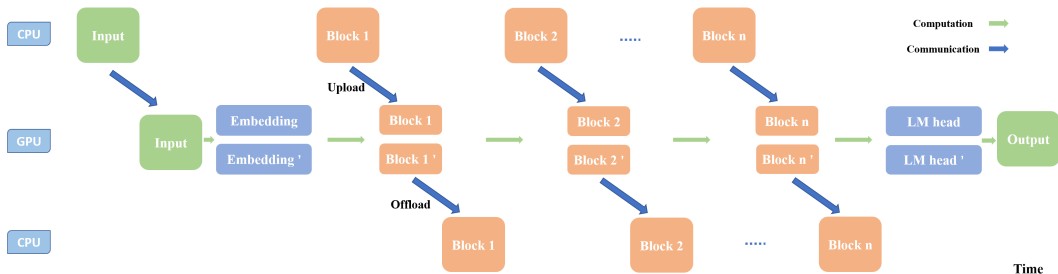

Figure 6: Workflow of the naive and non-overlap ZO-Offloading framework with only dual forward passes. This diagram demonstrates the sequential process without communication and computation overlap, using the pure PyTorch framework.

**Why the Dynamic Scheduler and Overlap Matter** Figure 6 provides a visual depiction of the workflow in the naive ZO-Offloading framework, specifically illustrating the naive, non-overlapping approach to dual forward passes. In this workflow, data is initially loaded from the CPU to the GPU, starting with the input processed through the embedding layer. Each transformer block (from Block 1 to Block n) is then sequentially processed: first uploaded to the GPU, where dual forward computations occur, and then offloaded back to the CPU after computation is complete.

This step-by-step process highlights a significant inefficiency in the current implementation: the GPU must wait for each block to be offloaded back to the CPU before the next block can be uploaded and processed. This results in substantial idle times for the GPU during offloads, and the CPU during uploads, as each unit must wait for the other to complete its task before proceeding. Such lack of overlap between computation (green arrows) and communication (blue arrows) tasks demonstrates a critical area for improvement, underlining the necessity for an overlapped or asynchronous approach to enhance overall system efficiency and throughput. By addressing this inefficiency, we can significantly reduce the training time and increase the utilization of both CPU and GPU resources.

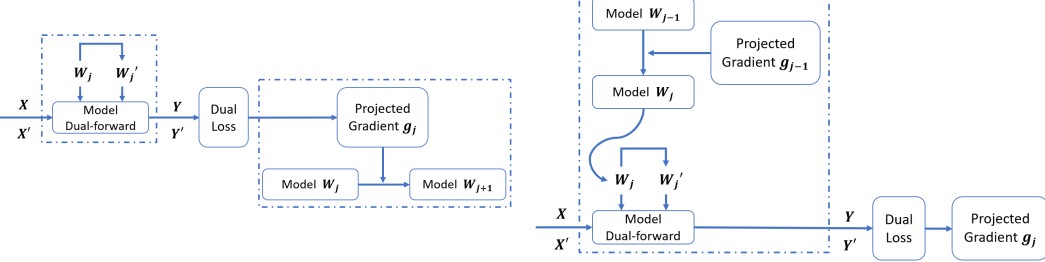

(a) Model parameter updates without the efficient strategy.

(b) Model parameter updates with the efficient strategy.

Figure 7: **Comparison of model parameters updates without/with efficient strategy**. (a) illustrates the process where, at the $j$-th iteration, the model computes the projected gradient $g_j$ using the dual-forward method and subsequently updates the model parameters. (b) demonstrates that at the $j$-th iteration, the model first updates the parameters using the previously saved projected gradient $g_{j-1}$, and then performs the dual-forward pass to compute the new projected gradient $g_j$.

Figure 7 illustrates the traditional and efficient approaches to model parameter updates. Typically, in the $j$-th iteration, model parameters are updated post the dual-forward passes, which necessitates the offloading of parameters from the GPU to the CPU following these computations. This offloading results in the parameters needing to be re-uploaded to the GPU solely for updates, leading to dual communication overhead (indicated by the two dotted boxes).

In contrast, our strategy reconfigures the $j$-th iteration by first applying the projected gradient from the $j-1$-th iteration to update the model parameters. Subsequently, the $j$-th dual-forward pass is performed to compute the new projected gradient. This adjustment reduces the communication demands to a single instance (indicated by the one dotted box) per iteration, streamlining the entire process and reducing time delays associated with multiple data transfers.

## C EXPERIMENT SETTINGS

1. MODEL SPECIFICATIONS:

- **Model Family:** We used the Open Pre-trained Transformer (OPT) (Zhang et al., 2022) model family for our experiments, ranging from 125 million to 175 billion parameters, to assess our framework's scalability and performance across different complexities.
- **Baseline Model:** The MeZO (Memory-efficient Zeroth-Order) serves as the baseline for comparison, known for its efficiency in memory throughput among Zeroth-Order offloading methods.

2. DATASET:

- **Dataset Used:** All performance evaluation experiments were conducted using the Stanford Sentiment Treebank (SST-2) dataset, a standard benchmark for evaluating natural language processing models.

3. HYPERPARAMETERS:

- **Learning Rate:** $1 \times 10^{-7}$
- **Steps:** 100
- **Batch Size:** 1
- **Sequence Length:** 2048

4. COMPUTATIONAL RESOURCES:

- **GPU:** NVIDIA A100 with 80GB of memory.
- **CPU:** AMD Milan.
- **Software:** Experiments were conducted using Python version 3.11, PyTorch 2.4.0, and CUDA 12.1.

5. EVALUATION METRICS:

- **GPU Memory Usage:** Measured in gigabytes (GiB).
- **Throughput:** Evaluated as tokens per second to assess the efficiency of the model training under various configurations.

## D MORE EXPERIMENT RESULTS

Table 4: **Main results of ZO-Offloading precision on OPT-13B**

| Method | SST-2 (%) | RTE (%) | CB (%) | BoolQ (%) | WSC (%) | WIC (%) | MultiRC (%) |
|---|---|---|---|---|---|---|---|
| MeZO | 91.4 | 66.1 | 67.9 | 67.6 | 63.5 | 61.1 | 60.1 |
| ZO-Offloading | 91.4 | 66.1 | 67.9 | 67.6 | 63.5 | 61.1 | 60.1 |

**Results on Accuracy.** In this experimental evaluation, we aim to demonstrate the effectiveness of the ZO-Offloading method in maintaining precision across multiple NLP benchmarks when fine-tuning the OPT-13B model. The benchmarks selected for this study include SST-2 (Socher et al., 2013) for sentiment analysis, RTE (Dagan et al., 2005) for recognizing textual entailment, CB (De Marneffe et al., 2019) for coreference resolution, BoolQ (Clark et al., 2019) for question answering, WSC (Levesque et al., 2012) for Winograd schema challenge, WIC (Pilehvar &

Camacho-Collados, 2018) for word-in-context disambiguation, and MultiRC (Khashabi et al., 2018) for multiple-choice reading comprehension. These datasets are chosen due to their diverse linguistic challenges and the depth of language understanding they require.

As shown in Table 4, ZO-Offloading achieves identical precision rates to the baseline MeZO approach across all evaluated benchmarks. This parity in performance is significant as it underscores the ZO-Offloading's ability to effectively maintain model precision despite the GPU memory usage reductions afforded by zeroth-order optimizers.

Table 5: **Throughput (token/sec) results to validate proposed features.**

| Model | MeZO | ZO-Offloading (no scheduler overlap) | ZO-Offloading (no reusable memory) | ZO-Offloading (no efficient update) | ZO-Offloading |
|---|---|---|---|---|---|
| OPT-125M | 14889 | 9486 (x0.64) | 5807 (x0.39) | 13031 (x0.88) | 13074 (x0.89) |
| OPT-350M | 5274 | 3432 (x0.65) | 1951 (x0.37) | 5099 (x0.97) | 5099 (x0.97) |
| OPT-1.3B | 1954 | 1109 (x0.57) | 735 (x0.38) | 1567 (x0.80) | 1954 (x1.00) |
| OPT-2.7B | 1087 | 573 (x0.52) | 422 (x0.39) | 849 (x0.78) | 1087 (x1.00) |
| OPT-6.7B | 499 | 225 (x0.45) | 184 (x0.37) | 373 (x0.74) | 499 (x1.00) |
| OPT-13B | 270 | 105 (x0.39) | 103 (x0.38) | 198 (x0.73) | 270 (x1.00) |
| OPT-30B | - | 35 | 46 | 81 | 122 |
| OPT-66B | - | 22 | 15 | 36 | 40 |
| OPT-175B | - | 8 | 5 | 13 | 14 |

**Full Ablation Study on Throughput.** This comprehensive ablation study extends our evaluation across the entire OPT model family, from 125 million to 175 billion parameters, validating the impact of key features on throughput. As detailed in Table 5, removing scheduler overlap consistently leads to notable throughput reductions, particularly in larger models, highlighting its importance in task management. The absence of reusable memory shows the most substantial decreases across all sizes, emphasizing its role in efficient memory management. Similarly, disabling efficient parameter updating variably impacts throughput, with larger models demonstrating a critical dependence on this feature for maintaining performance.

Table 6: **Throughput (token/sec) results to validate proposed asynchronous checkpointing.**

| Model | MeZO | MeZO (torch.save) | ZO-Offloading | ZO-Offloading (torch.save) | ZO-Offloading (Async-Checkpoint) |
|---|---|---|---|---|---|
| OPT-1.3B | 1954 | 319 (x0.16) | 1954 (x1.00) | 462 (x0.24) | 1954 (x1.00) |
| OPT-2.7B | 1087 | 160 (x0.15) | 1087 (x1.00) | 221 (x0.20) | 1087 (x1.00) |
| OPT-6.7B | 499 | 52 (x0.10) | 499 (x1.00) | 88 (x0.18) | 499 (x1.00) |

**Asynchronous Checkpointing Experiment and Results Analysis.** The asynchronous checkpointing feature was implemented in our ZO-Offloading framework to minimize the delays associated with traditional checkpointing in large-scale models like OPT-1.3B, OPT-2.7B, and OPT-6.7B. The experiment tested five scenarios: MeZO without checkpointing ("MeZO"), MeZO with traditional synchronous checkpointing using `torch.save` ("MeZO (torch.save)"), ZO-Offloading without checkpointing ("ZO-Offloading"), ZO-Offloading with traditional synchronous checkpointing using `torch.save` ("ZO-Offloading (torch.save)"), and ZO-Offloading with asynchronous checkpointing ("ZO-Offloading (Async-Checkpoint)"). The checkpointing process involved dividing model parameters into two halves, $p1$ and $p2$, which were alternately saved to disk asynchronously to prevent interruption in model computation.

The throughput results, detailed in Table 6, show that traditional checkpointing with `torch.save()` significantly reduces throughput across all models tested, with the most considerable drop seen in the OPT-6.7B model to just 10% of its baseline performance. However, the drop is less severe in "ZO-Offloading (torch.save)" compared with "MeZO (torch.save)" due to the limited data transfer time from GPU to CPU. We can see that ZO-Offloading with asynchronous checkpointing maintained full baseline throughput compared with ZO-Offloading without checkpointing. These findings demonstrate the effectiveness of the asynchronous checkpointing mechanism, which ensures that the training process remains uninterrupted and efficient.

**Differential Batch-size and Sequence Length Analysis.** This analysis explores the impact of varying batch sizes and sequence lengths on the performance of the ZO-Offloading compared to

Table 7: **Different batch-size analysis.**

| Model | B | Memory Usage (MB) | | Throughput (tokens/sec) | |
|---|---|---|---|---|---|
| | | MeZO | ZO-Offloading | MeZO | ZO-Offloading |
| OPT-1.3B | | 9117 | 4413 (x0.48) | 1954 | 1954 (x1.00) |
| OPT-2.7B | 1 | 15277 | 5261 (x0.34) | 1087 | 1087 (x1.00) |
| OPT-6.7B | | 32083 | 8329 (x0.26) | 499 | 499 (x1.00) |
| OPT-1.3B | | 10809 | 6617 (x0.61) | 1055 | 1055 (x1.00) |
| OPT-2.7B | 2 | 16575 | 7563 (x0.46) | 594 | 594 (x1.00) |
| OPT-6.7B | | 33857 | 9865 (x0.29) | 278 | 278 (x1.00) |
| OPT-1.3B | | 13249 | 9451 (x0.71) | 566 | 566 (x1.00) |
| OPT-2.7B | 4 | 19409 | 10397 (x0.54) | 312 | 312 (x1.00) |
| OPT-6.7B | | 37239 | 13485 (x0.36) | 145 | 145 (x1.00) |
| OPT-1.3B | | 18917 | 15119 (x0.80) | 289 | 289 (x1.00) |
| OPT-2.7B | 8 | 24745 | 16065 (x0.65) | 160 | 160 (x1.00) |
| OPT-6.7B | | 42278 | 19153 (x0.45) | 75 | 75 (x1.00) |

Table 8: **Different sequence length analysis.**

| Model | Length | Memory Usage (MB) | | Throughput (tokens/sec) | |
|---|---|---|---|---|---|
| | | MeZO | ZO-Offloading | MeZO | ZO-Offloading |
| OPT-1.3B | | 8333 | 3747 (x0.45) | 3689 | 3689 (x1.00) |
| OPT-2.7B | 1024 | 14175 | 4669 (x0.33) | 2092 | 2092 (x1.00) |
| OPT-6.7B | | 31475 | 7721 (x0.25) | 901 | 901 (x1.00) |
| OPT-1.3B | | 9117 | 4413 (x0.48) | 1954 | 1954 (x1.00) |
| OPT-2.7B | 2048 | 15277 | 5261 (x0.34) | 1087 | 1087 (x1.00) |
| OPT-6.7B | | 32083 | 8329 (x0.26) | 499 | 499 (x1.00) |
| OPT-1.3B | | 11379 | 7581 (x0.67) | 830 | 830 (x1.00) |
| OPT-2.7B | 4096 | 16973 | 8453 (x0.50) | 490 | 490 (x1.00) |
| OPT-6.7B | | 35549 | 11319 (x0.32) | 250 | 250 (x1.00) |
| OPT-1.3B | | 32051 | 28253 (x0.88) | 302 | 302 (x1.00) |
| OPT-2.7B | 8192 | 37693 | 29173 (x0.77) | 187 | 187 (x1.00) |
| OPT-6.7B | | 54365 | 32183 (x0.59) | 108 | 108 (x1.00) |

the MeZO baseline. Tables 7 and 8 present the memory usage and throughput metrics for different configurations of the OPT models, ranging from 1.3B to 6.7B parameters. Table 7 shows the results for different batch-sizes. As batch size increases, there is a consistent trend where ZO-Offloading maintains throughput equivalency with MeZO across all model sizes, despite significant reductions in memory usage. Even at higher batch sizes, ZO-Offloading demonstrates robust performance, showing no decrease in throughput relative to its MeZO counterpart. For example, in the OPT-1.3B model at a batch size of 8, the throughput remains constant at 289 tokens/sec, maintaining operational efficiency irrespective of the increased computational load.

Table 8 illustrates the impact of sequence length on throughput. Similar to the batch-size analysis, increasing the sequence length does not compromise the throughput of ZO-Offloading, maintaining parity with the MeZO model across varying lengths. Notably, even at a sequence length of 8192 for the OPT-1.3B model, ZO-Offloading sustains a throughput of 302 tokens/sec, effectively handling larger input sizes without a drop in performance. The analyses confirm that ZO-Offloading effectively manages larger batch sizes and sequence lengths without sacrificing throughput. This resilience is crucial for practical deployments where varying input sizes and batch configurations are common, underscoring the scalability and robustness of the ZO-Offloading approach in diverse operational environments.

Table 9: **Complete throughput (token/sec) results to validate AMP Mode.** AMP auto-cast with FP16 (top) and BF16 (below).

| Model | ZO-Offload (non-compress) | ZO-Offload (FP16) | ZO-Offload (BF16) | ZO-Offload (FP8) |
|---|---|---|---|---|
| OPT-1.3B | 4827 | 4770 (x0.988) | 4760 (x0.986) | 4802 (x0.995) |
| OPT-2.7B | 2811 | 2974 (x1.058) | 2974 (x1.058) | 2997 (x1.066) |
| OPT-6.7B | 1271 | 1641 (x1.291) | 1641 (x1.291) | 1662 (x1.308) |
| OPT-13B | 561 | 930 (x1.658) | 930 (x1.658) | 951 (x1.695) |
| OPT-30B | 286 | 416 (x1.455) | 416 (x1.455) | 425 (x1.486) |
| OPT-66B | 127 | 192 (x1.512) | 192 (x1.512) | 198 (x1.559) |
| OPT-175B | 43 | 65 (x1.512) | 65 (x1.512) | 68 (x1.584) |
| OPT-1.3B | 4565 | 4430 (x0.970) | 4430 (x0.970) | 4463 (x0.978) |
| OPT-2.7B | 2778 | 2816 (x1.014) | 2816 (x1.014) | 2818 (x1.014) |
| OPT-6.7B | 1273 | 1594 (x1.252) | 1594 (x1.252) | 1612 (x1.266) |
| OPT-13B | 678 | 910 (x1.342) | 910 (x1.342) | 924 (x1.363) |
| OPT-30B | 285 | 407 (x1.428) | 407 (x1.428) | 415 (x1.456) |
| OPT-66B | 127 | 188 (x1.480) | 188 (x1.480) | 194 (x1.528) |
| OPT-175B | 43 | 64 (x1.488) | 64 (x1.488) | 67 (x1.565) |

Table 10: **More Experiment Results for BLOOM (Workshop et al., 2023).** Instances of '-' in the table indicate scenarios where the corresponding method failed to execute due to memory constraints. The values in parentheses (x) represent the ratio of each measurement compared to the baseline MeZO (first column) configuration.

| Model | GPU Memory Usage (MB) ↓ | | | | Throughput (tokens/sec) ↑ | | | |
|---|---|---|---|---|---|---|---|---|
| | MeZO(32) | ZO-Offload(32) | MeZO(16) | ZO-Offload(16) | MeZO(32) | ZO-Offload(32) | MeZO(16) | ZO-Offload(16) |
| BLOOM-176B | - | 49525 | - | **24864** | - | 14 | - | **37** |