# OpenReview forum: "ZO-Offloading: Fine-Tuning LLMs with 100 Billion Parameters on a Single GPU"
_ICLR.cc/2025/Conference — ICLR 2025 Conference Withdrawn Submission_

### Official Review · Reviewer_tDhK · 2024-10-27

**Soundness:** 1
**Presentation:** 2
**Contribution:** 2
**Rating:** 1
**Confidence:** 4

**Summary:**

This paper present to conduce CPU offloading for zero-order optimizer methods in LLM fine-tuning/post-training. The problem to solve is practical and useful. The implementation incorporating with Nvidia's Automatic Mixed Precision (AMP) is necessary and essential. The experiment results show good speedup compared with previous zero-order method as MeZO.

**Strengths:**

Paper presentation is good and logic flow is clear.

Problem to solve is a good direction.

Auto-cast and AMP support is a good thing to have for CPU offloading scenarios.

**Weaknesses:**

My first major concern is, this paper seems not technical sound and may have misunderstanding in cuda and amp techniques:

In Figure 1, it is unfair and unreasonable to compare memory consumption of using Adam vs using SGD, since Adam offers higher model training quality in general compared with simple SGD. That is the reason why Adam is dominantly adopted in LLM world. Comparing optimizer without aligning on model quality is unfair. Please provide more comparison results on different optimizers and its convergence results

Additionally, how did the authors collect the memory usage size numbers for different optimizers on a 24GB mem size GPU? If empirically measured, how to measure memory usage size (e.g.,68870MB roughly 68GB) over GPU memory capacity (24GB)?


Also the paper's implementation is not practical and not sound. from line 295 to 305 on page 6, the authors describe sync between computation and offload H2D D2H mem data transfer is implemented by the authors with some **lock** mechanism. This is impractical and cuda already provided multiple data transfer and compute sync techniques. For example, for memcpy specifically, it can be async or sync, by forcing it to be synchronized memcpy, there is no need at all to use any extra **lock** developed by the authors. For more generally multi-stream sync, cuda offers plenty of synchronization methods at stream/event/block/thread level. Usually, it is unnecessary to build new wheels without leveraging existing more efficient functionality. Please illustrate more on why a customized lock mechanism is needed here.

Further, by looking at supplementary materials, on page 16 line 812, the author reports using pytorch **3.11**. As far as I know, pytorch has not have any **3.x** version yet. I think authors may not familiar with the basic framework as well. Even assuming it is a typo, the only pytorch version cover number **11** is version **1.11.0**, which is quite old and experiment numbers based on this pytorch version seems a bit outdated and unconvincing. Please discuss more on which pytorch version is selected and why.

My second concern is about paper motivation, zero-order methods are not widely used for real large model training, as it is widely agreed these kind of gradient estimation methods could lead to model divergence. Please provide more citations or example applications on how zero-order methods is adopted in real world model training.


My third concern is paper novelty. Overall the paper's system design and implementation are very similar to zero-offload (https://arxiv.org/pdf/2101.06840) case (e.g. overlap data memcpy with computation as sec 5.1, dedicated memory block reuse and mem management as sec 5.2, AMP support as sec 5.4 which is by default supported in zero-offload code inside deepspeed).

Although sec 5.5 briefly discussed extension to async checkpointing seems novel, it mentioned async checkpoint without interfere training pipeline, this kind of idea already have much more solid design and implementation such as checkfreq, check-n-run

CheckFreq: Frequent, Fine-Grained DNN Checkpointing, FAST'21

Check-N-Run: a Checkpointing System for Training Deep Learning Recommendation Models, NSDI'22

Please provide a more detailed comparison of this zo-offload approach with zero-offload and other related works, highlighting specific novel aspects of this method.

My fourth concern is evaluation, it only reports simple throughput or token per sec results, without reports any model convergence/accuracy tests compared with more widely adopted first order methods. Could you include convergence comparison with first order methods?

some minor issues:
1. for figure 5a 5b mentioned in introduction, it would be great to cover them in main text Not appendix because these figures are essential for reader to understand main difference between first order and zero order methods.
2. Figure 3 and 4 seems 90% identical, maybe merge into 1 figure and highlight which part is new for AMP support.
3. For a more uniformed paper structure and shape, it is better to only describe and illustrate a few major contributions rather than extending methods to multiple minor ones (e.g. extending to async checkpoint).

**Questions:**

1. regarding to paper novelty, what big/major LLM model is pre-train/post-train using zero-order optimization methods?

2. For system design and implementation, how it different from zero-offload?

3. how higher memory usage (e.g, 68870MB) profiling is conducted with 24GB memory GPU?

---

> ### Author Response · Authors · 2024-11-26
> **Response by Authors (Part1)**
>
> Thank you for your detailed feedback. We appreciate the opportunity to address your concerns and provide further clarity on our contributions.
>
> Response to Your First Concern
>
> > My first major concern is, this paper seems not technical sound and may have misunderstanding in cuda and amp techniques:
>
> > In Figure 1, it is unfair and unreasonable to compare memory consumption of using Adam vs using SGD, since Adam offers higher model training quality in general compared with simple SGD. That is the reason why Adam is dominantly adopted in LLM world. Comparing optimizer without aligning on model quality is unfair. Please provide more comparison results on different optimizers and its convergence results
>
> > Additionally, how did the authors collect the memory usage size numbers for different optimizers on a 24GB mem size GPU? If empirically measured, how to measure memory usage size (e.g.,68870MB roughly 68GB) over GPU memory capacity (24GB)?
>
> > Also the paper's implementation is not practical and not sound. from line 295 to 305 on page 6, the authors describe sync between computation and offload H2D D2H mem data transfer is implemented by the authors with some lock mechanism. This is impractical and cuda already provided multiple data transfer and compute sync techniques. For example, for memcpy specifically, it can be async or sync, by forcing it to be synchronized memcpy, there is no need at all to use any extra lock developed by the authors. For more generally multi-stream sync, cuda offers plenty of synchronization methods at stream/event/block/thread level. Usually, it is unnecessary to build new wheels without leveraging existing more efficient functionality. Please illustrate more on why a customized lock mechanism is needed here.
>
> > Further, by looking at supplementary materials, on page 16 line 812, the author reports using pytorch 3.11. As far as I know, pytorch has not have any 3.x version yet. I think authors may not familiar with the basic framework as well. Even assuming it is a typo, the only pytorch version cover number 11 is version 1.11.0, which is quite old and experiment numbers based on this pytorch version seems a bit outdated and unconvincing. Please discuss more on which pytorch version is selected and why.
>
> 1. We acknowledge the trade-off between first-order optimizers, which require more memory but generally offer higher precision, and zeroth-order optimizers. However, as demonstrated in [MeZO](https://arxiv.org/abs/2305.17333), zeroth-order optimizers can achieve almost the same precision as first-order optimizers for fine-tuning OPT-13B, making them a viable alternative in memory-constrained settings. Building on this, we believe our framework has the potential to enable the community to fine-tune even larger models, such as OPT-175B, further advancing the applicability of zeroth-order optimization for large-scale model fine-tuning.
>
> 2. While CUDA provides robust synchronization methods for coordinating streams, our locking mechanism addresses a distinct issue related to PyTorch's memory management, rather than focusing solely on operation scheduling or synchronization. PyTorch’s memory pool reuses tensor memory once the reference count drops to zero and the tensor is collected by Python’s garbage collector. In the context of **multiple asynchronous streams**, memory conflicts can arise if the memory of a tensor is reused for a new allocation before all operations on the original tensor are fully completed. Our locking mechanism ensures that memory is reused only after all associated operations have been finalized, preventing conflicts caused by premature memory reuse. This approach is specifically tailored to address challenges related to memory safety in multi-stream environments and complements existing synchronization techniques provided by CUDA.
>
> 3. We clarified in the appendix that the GPU used in our experiments was an NVIDIA A100 with 80GB of memory. While we limited GPU memory usage to 24GB for our ZO-offloading framework, we did not use a GPU with only 24GB of capacity. Regarding the versioning issue, the “3.11” mentioned in the paper refers to the Python version. The PyTorch version we used is 2.4.0. We apologize for any confusion caused by this oversight.

---

> > ### Comment · Reviewer_tDhK · 2024-11-27
> > **Replay by Reviewer (part1)**
> >
> > Thanks authors for the reply.
> >
> > >1 We acknowledge the trade-off between first-order optimizers, which require more memory but generally offer higher precision, and zeroth-order optimizers. However, as demonstrated in MeZO, zeroth-order optimizers can achieve almost the same precision as first-order optimizers for fine-tuning OPT-13B, making them a viable alternative in memory-constrained settings. Building on this, we believe our framework has the potential to enable the community to fine-tune even larger models, such as OPT-175B, further advancing the applicability of zeroth-order optimization for large-scale model fine-tuning.
> >
> > It is general accepted and common practice that adam achieve higher accuracy than sgd. I do not think a single/few paper could make this statement false.
> >
> > >2 While CUDA provides robust synchronization methods for coordinating streams, our locking mechanism addresses a distinct issue related to PyTorch's memory management, rather than focusing solely on operation scheduling or synchronization. PyTorch’s memory pool reuses tensor memory once the reference count drops to zero and the tensor is collected by Python’s garbage collector. In the context of multiple asynchronous streams, memory conflicts can arise if the memory of a tensor is reused for a new allocation before all operations on the original tensor are fully completed. Our locking mechanism ensures that memory is reused only after all associated operations have been finalized, preventing conflicts caused by premature memory reuse. This approach is specifically tailored to address challenges related to memory safety in multi-stream environments and complements existing synchronization techniques provided by CUDA.
> >
> > As you said, pytorch/python GC already reuse tensor when no reference, and I don't see why it is unsafe. Cuda streams are always async, and I never hear of major memory leak issue caused by these GC. And I don't see how your lock mechanism is better than pytorch GC. Feel free to provide your code piece gist and illustrate how it is better than torch GC.
> >
> > > 3 We clarified in the appendix that the GPU used in our experiments was an NVIDIA A100 with 80GB of memory. While we limited GPU memory usage to 24GB for our ZO-offloading framework, we did not use a GPU with only 24GB of capacity. Regarding the versioning issue, the “3.11” mentioned in the paper refers to the Python version. The PyTorch version we used is 2.4.0. We apologize for any confusion caused by this oversight.
> >
> > Thanks for the correction. Please correct the text in your introduction Line 103- 104 bold font text and appendix as well.

---

> ### Author Response · Authors · 2024-11-26
> **Response by Authors (Part2)**
>
> For Your Second Concern
>
> > My second concern is about paper motivation, zero-order methods are not widely used for real large model training, as it is widely agreed these kind of gradient estimation methods could lead to model divergence. Please provide more citations or example applications on how zero-order methods is adopted in real world model training.
>
> As mentioned earlier, [MeZO](https://arxiv.org/abs/2305.17333), a highly cited work, has demonstrated that zeroth-order optimizers can maintain nearly the same precision as first-order methods for fine-tuning OPT-13B. This provides strong evidence of the viability of zeroth-order methods for real-world applications. More studies like [[1](https://arxiv.org/abs/2404.08080)], [[2](https://arxiv.org/abs/2402.11592)], [[3](https://arxiv.org/abs/2312.15184)], [[4](https://arxiv.org/abs/2410.08989)], [[5](https://arxiv.org/abs/2401.04343)], and [[6](https://arxiv.org/abs/2310.09639)].
>
> [1] https://arxiv.org/abs/2404.08080 \
> [2] https://arxiv.org/abs/2402.11592 \
> [3] https://arxiv.org/abs/2312.15184 \
> [4] https://arxiv.org/abs/2410.08989 \
> [5] https://arxiv.org/abs/2401.04343 \
> [6] https://arxiv.org/abs/2310.09639
>
> For Your Third Concern
>
> > My third concern is paper novelty. Overall the paper's system design and implementation are very similar to zero-offload (https://arxiv.org/pdf/2101.06840) case (e.g. overlap data memcpy with computation as sec 5.1, dedicated memory block reuse and mem management as sec 5.2, AMP support as sec 5.4 which is by default supported in zero-offload code inside deepspeed).
>
> > Although sec 5.5 briefly discussed extension to async checkpointing seems novel, it mentioned async checkpoint without interfere training pipeline, this kind of idea already have much more solid design and implementation such as checkfreq, check-n-run
>
> > CheckFreq: Frequent, Fine-Grained DNN Checkpointing, FAST'21
>
> > Check-N-Run: a Checkpointing System for Training Deep Learning Recommendation Models, NSDI'22
>
> > Please provide a more detailed comparison of this zo-offload approach with zero-offload and other related works, highlighting specific novel aspects of this method.
>
> 1. The primary contribution of our work lies in the seamless integration of CPU offloading with the zeroth-order optimizer. The introduction clarifies the differences between first-order and zeroth-order methods and explains why zeroth-order methods are more suitable for parameter offloading to CPU than first-order methods.
>
> Regarding the differences between Zero-Offload and our ZO-Offloading, while the names may appear similar, the offloading strategies are fundamentally different. Zero-Offload offloads the **whole model**'s gradients, first-order optimizer states, and parameter update computations to the CPU, while keeping the entire model parameters and forward/backward computations on the GPU. In contrast, ZO-Offloading divides transformer parameters into **blocks** and processes them streamingly: uploading blocks from the CPU to GPU, performing computations on the GPU, and offloading them back to the CPU **block by block**. This distinct offloading strategy necessitates entirely different memory management, scheduling, overlap, and AMP designs, tailored specifically to support the zeroth-order optimization approach.
>
> 2. Many existing frameworks, including Megatron, DeepSpeed, and even vanilla PyTorch, incorporate technologies such as memory pooling, computation-communication overlap, and AMP. Our work does not aim to reinvent these technologies but instead focuses on how they are adapted and applied to address the unique challenges of zeroth-order optimizers.
>
> 3. We appreciate the references to CheckFreq and Check-N-Run, which indeed offer solid designs for asynchronous checkpointing. However, we would like to clarify that the primary contributions of our paper focus on memory and speed optimizations specifically tailored for the zeroth-order optimizer. The discussion of asynchronous checkpointing in Section 5.5 is included as a potential extension and is not a central focus of this work. It is presented to illustrate a possible avenue for future exploration rather than as a fully developed contribution. We believe this aspect is beyond the main discussion scope of this paper, and we hope this clarifies its role within the context of our work.
>
> For Your Fourth Concern
>
> > My fourth concern is evaluation, it only reports simple throughput or token per sec results, without reports any model convergence/accuracy tests compared with more widely adopted first order methods. Could you include convergence comparison with first order methods?
>
> Our framework maintains the same precision as [MeZO](https://arxiv.org/abs/2305.17333), which has already provided accuracy evaluations against first-order methods. As such, we believe it is unnecessary to replicate those results in this paper, as the focus here is on memory and speed optimizations rather than a comparative study of convergence.

---

> > ### Comment · Reviewer_tDhK · 2024-11-27
> > **Reply By Reviewer (part2)**
> >
> > > As mentioned earlier, MeZO, a highly cited work, has demonstrated that zeroth-order optimizers can maintain nearly the same precision as first-order methods for fine-tuning OPT-13B. This provides strong evidence of the viability of zeroth-order methods for real-world applications. More studies like [1], [2], [3], [4], [5], and [6].
> >
> > Again, It is general accepted and common practice that adam achieve higher accuracy than sgd. I don't think a few paper could make this statement false.
> > >1 The primary contribution of our work lies in the seamless integration of CPU offloading with the zeroth-order optimizer. The introduction clarifies the differences between first-order and zeroth-order methods and explains why zeroth-order methods are more suitable for parameter offloading to CPU than first-order methods.
> >
> > The main contribution of this paper is as in your above comment: "the focus here is on memory and speed optimizations". Therefore if nothing is novel on memory/speed optimization, then the paper novelty is limited since it is just a new workload using zero-offload.
> >
> > > Regarding the differences between Zero-Offload and our ZO-Offloading, while the names may appear similar, the offloading strategies are fundamentally different. Zero-Offload offloads the whole model's gradients, first-order optimizer states, and parameter update computations to the CPU, while keeping the entire model parameters and forward/backward computations on the GPU. In contrast, ZO-Offloading divides transformer parameters into blocks and processes them streamingly: uploading blocks from the CPU to GPU, performing computations on the GPU, and offloading them back to the CPU block by block. This distinct offloading strategy necessitates entirely different memory management, scheduling, overlap, and AMP designs, tailored specifically to support the zeroth-order optimization approach.
> >
> > Please read ZeRO-Offload paper and ZeRO-Infinity paper (e.g. https://arxiv.org/pdf/2104.07857 , Figure 4 notation last line, " $𝑃_2^0$ is the portion of layer 0’s parameters") from deepspeed and also take a look of code from deepspeed repo. **block by block** offload is standard practice in CPU offloading starting from 2020. This response above just indicates the authors may not be very familiar with existed related work.
> >
> > >2 Many existing frameworks, including Megatron, DeepSpeed, and even vanilla PyTorch, incorporate technologies such as memory pooling, computation-communication overlap, and AMP. Our work does not aim to reinvent these technologies but instead focuses on how they are adapted and applied to address the unique challenges of zeroth-order optimizers.
> >
> > Novelty is limited if just using pytorch/deepspeed/megatron offload technology with new input (zero-order) tensor.
> >
> > >3 We appreciate the references to CheckFreq and Check-N-Run, which indeed offer solid designs for asynchronous checkpointing. However, we would like to clarify that the primary contributions of our paper focus on memory and speed optimizations specifically tailored for the zeroth-order optimizer. The discussion of asynchronous checkpointing in Section 5.5 is included as a potential extension and is not a central focus of this work. It is presented to illustrate a possible avenue for future exploration rather than as a fully developed contribution. We believe this aspect is beyond the main discussion scope of this paper, and we hope this clarifies its role within the context of our work.
> >
> > If it is irrelevant and not solid, please exclude it.
> >
> > >Our framework maintains the same precision as MeZO, which has already provided accuracy evaluations against first-order methods. As such, we believe it is unnecessary to replicate those results in this paper, as the focus here is on memory and speed optimizations rather than a comparative study of convergence.
> >
> > We believe it is necessary to have convergence comparison with first-order method on LLama 3.1 or GPT at least to verify whether it can be generally applied on widely adopted LLMs.

---

> ### Author Response · Authors · 2024-11-26
> **Response by Authors (Part3)**
>
> Additional Responses to Minor Issues
>
> > for figure 5a 5b mentioned in introduction, it would be great to cover them in main text Not appendix because these figures are essential for reader to understand main difference between first order and zero order methods.
>
> We believe it is important to maintain a concise introduction, therefore, we have provided a brief description of our motivation in the main text and included a more detailed explanation in the Appendix.
>
> > Figure 3 and 4 seems 90% identical, maybe merge into 1 figure and highlight which part is new for AMP support.
>
> We believe it is essential to keep Figure 2 concise, therefore, we have summarized the AMP component in a separate Figure 3. While Figure 2 primarily illustrates our Offloading strategy, Figure 3 focuses on how AMP is integrated into this strategy. This separation underscores a recursive concept, enhancing the reader's understanding of our core contributions.
>
> > For a more uniformed paper structure and shape, it is better to only describe and illustrate a few major contributions rather than extending methods to multiple minor ones (e.g. extending to async checkpoint).
>
> We appreciate your suggestion to streamline the structure by focusing on major contributions. However, in the original version, we have explicitly marked the asynchronous checkpointing section as an "extension" in the manuscript to indicate that it is not a primary contribution but rather a potential avenue for future exploration.
>
> For your questions, we want to claim again:
>
> > regarding to paper novelty, what big/major LLM model is pre-train/post-train using zero-order optimization methods?
>
> [MeZO](https://arxiv.org/abs/2305.17333) demonstrates that fine-tuning OPT-13B using only the zeroth-order method can achieve precision comparable to first-order methods. Building on this, we believe our framework has the potential to enable the community to fine-tune even larger models, such as OPT-175B, further advancing the applicability of zeroth-order optimization for large-scale model fine-tuning.
>
> > For system design and implementation, how it different from zero-offload?
>
> The primary contribution of our work lies in the seamless integration of CPU offloading with the zeroth-order optimizer. The introduction clarifies the differences between first-order and zeroth-order methods and explains why zeroth-order methods are more suitable for parameter offloading to CPU than first-order methods.
> Regarding the differences between Zero-Offload and our ZO-Offloading, while the names may appear similar, the offloading strategies are fundamentally different. Zero-Offload offloads the **whole model**'s gradients, first-order optimizer states, and parameter update computations to the CPU, while keeping the entire model parameters and forward/backward computations on the GPU. In contrast, ZO-Offloading divides transformer parameters into **blocks** and processes them streamingly: uploading blocks from the CPU to GPU, performing computations on the GPU, and offloading them back to the CPU **block by block**. This distinct offloading strategy necessitates entirely different memory management, scheduling, overlap, and AMP designs, tailored specifically to support the zeroth-order optimization approach.
>
> > how higher memory usage (e.g, 68870MB) profiling is conducted with 24GB memory GPU?
>
> We clarified in the appendix that the GPU used in our experiments was an NVIDIA A100 with 80GB of memory. While we limit GPU memory usage to 24GB for our ZO-offloading framework, we did not use a GPU with only 24GB of capacity.

---

> > ### Comment · Reviewer_tDhK · 2024-11-27
> > **Reply by Reviewer (part3)**
> >
> > Thanks for the responces here. I already replied major points in the other part1 and part2 replies. Thank you.

---

> ### Author Response · Authors · 2024-11-26
> **Request for Futher Feedback and Score Raise**
>
> Dear reviewer, today is the last day to revise the manuscript. If your concerns have been resolved, I kindly ask you to consider **raising your score**. If not, feel free to share any remaining questions. Thank you!

---

> > ### Comment · Reviewer_tDhK · 2024-11-27
> > **Reply on further feedback and score raise**
> >
> > Thanks again for the author's feedback.
> >
> > However, most of my concerns still not addressed, thus I remain my score. Thank you

---

### Official Review · Reviewer_JAuZ · 2024-10-29

**Soundness:** 2
**Presentation:** 2
**Contribution:** 2
**Rating:** 3
**Confidence:** 4

**Summary:**

The main goal of this paper is to allow fine-tuning of very large LLMs even on a single-GPU.  Their technique utilizes the host (CPU) memory for shuttling data between GPU and CPU memory. It maximizes GPU utilization by dynamically offloading model parameters to the CPU.
Parameter updates are integrated with ZO’s dual forward passes to minimize redundant data transfers. They have shown the integration of their technique with low-precision format. They claim to have no overheads compared to standard ZO methodologies.
They claim to be the first work that allows fine-tuning extremely large models, such as the OPT-175B with over 175 billion parameters, on a single GPU with just 24GB of memory.

**Strengths:**

They claim to be the first work that allows fine-tuning extremely large models, such as the OPT-175B with over 175 billion parameters, on a single GPU with just 24GB of memory. If so, this is a great feat and very useful to the community.
They utilize ZO's architectural features for CPU offloading, so that is an intelligent integration.

They have shown the integration of their technique with low-precision format.
Good ablation studies: they show breakup of performance due to several individual ideas.

**Weaknesses:**

* Some other works also claim to run OPT-175B on a single GPU, e.g.,

--FlexGen: High-Throughput Generative Inference of Large Language Models with a Single GPU
--LUT-GEMM: QUANTIZED MATRIX MULTIPLICATION BASED ON LUTS FOR EFFICIENT INFERENCE IN LARGE-SCALE GENERATIVE LANGUAGE MODELS
--OPTQ: ACCURATE POST-TRAINING QUANTIZATION FOR GENERATIVE PRE-TRAINED TRANSFORMERS

Please comment on them. If required, you may need to modify the sentence in your manuscript "With ZO-Offloading, for the first time, it becomes possible to fine-tune extremely large models...."


* Can you comment how your technique compares against
https://huggingface.co/docs/accelerate/index

* BLOOM models are also open-source. Running only OPT models limits your showcase of the applicability of this model. Communication delays are not primary bottleneck for OPT, but may be so for other LLMs.
Also, MeZO paper has shown results with multiple datasets; this paper shows results with only one dataset.
This reviewer has carefully examined the supplementary material also.

* Table 3: no benefit from using FP8, compared to FP16? Even FP16/BF16 are useful only for OPT-6.7B. This means that uploading/offloading low-precision data does not help much, which means the CPU-GPU transfer (Communication) is not a bottleneck. Actually, the real benefit of your technique will be clear when this communication is a bottleneck.

* Experimentation is somewhat weak and the proposed ideas are not very novel. Many works have already been done on shuttling data between CPU and GPU memories.


Minor:
* It may have been better to conduct ablation experiments on different CPU-GPU interconnect (PCIe) bandwidths/configurations, although the reviewer understands it is not easy. But some ablation could have been performed on GPU systems space.

* Figures 2 and 3 and 4 use white color for text, which gets faded (with color background) when printed.

* Figure 1, instead of 0, X should have been shown. 0 is misleading.
Figure 1 caption could mention that these values are observed for a single GPU, because with multiple GPUs, you can definitely find these numbers.

Figure 1 shows GPU memory or CPU memory?

* "previously unattainable with conventional methods" is redundant.

* Incomplete phrase "Since CPU resources can be combined and offloaded to expand the memory and computational capacity of a single GPU"

* Can you theoretically say: What is the largest OPT model that your technique can run on a 24GB GPU, assuming that any OPT size model is available?

* Table 3 should have been shown all the way till OPT 175B.

* Is ZO same as ZO-SGD? You are using both the terms.
* Comparison could have been performed with other techniques, if possible.

**Questions:**

See the comments above in the weakness section.

---

> ### Author Response · Authors · 2024-11-26
> **Response by Authors (Part1)**
>
> We believe the reviewer misunderstood our contributions. We appreciate the opportunity to clarify the distinctions between our work and the related works mentioned.
>
> 1. On FlexGen, LUT-GEMM, and OPTQ
>
> > Some other works also claim to run OPT-175B on a single GPU, e.g., --FlexGen: High-Throughput Generative Inference of Large Language Models with a Single GPU --LUT-GEMM: QUANTIZED MATRIX MULTIPLICATION BASED ON LUTS FOR EFFICIENT INFERENCE IN LARGE-SCALE GENERATIVE LANGUAGE MODELS --OPTQ: ACCURATE POST-TRAINING QUANTIZATION FOR GENERATIVE PRE-TRAINED TRANSFORMERS
> > Please comment on them. If required, you may need to modify the sentence in your manuscript "With ZO-Offloading, for the first time, it becomes possible to fine-tune extremely large models...."
>
> The works you referenced (FlexGen, LUT-GEMM, OPTQ) are **inference** frameworks designed to optimize tasks such as prefill, decode, and KV-cache management. In contrast, our work, ZO-Offloading, focuses on **fully fine-tuning** large language models. Fine-tuning requires fundamentally different techniques as it involves dual-forward passes with parameter perturbations, gradient computation, and parameter updates. The objectives and computational workflows of fine-tuning and inference are distinct, which is why a direct comparison with inference frameworks is less relevant.
>
> 2. Accelerate Library
>
> > Can you comment how your technique compares against https://huggingface.co/docs/accelerate/index
>
> Accelerate primarily targets distributed multi-GPU fine-tuning rather than single-GPU setups. While Accelerate does include a CPU offloading component, it is exclusively designed for first-order optimizers, incorporating gradient, activation value, and optimizer state offloading. This configuration is fundamentally incompatible with the ZO method, which does not utilize activation values or optimizer states and performs gradient calculations in-place. Consequently, the Accelerate framework cannot support offloading and scheduling for the ZO method. Moreover, Accelerate lacks a high-performance scheduler capable of optimizing the overlap between computation and communication.
>
> Our ZO-Offloading approach is specifically designed for the offloading of model parameters and their scheduling. ZO-Offloading divides transformer parameters into **blocks** and processes them streamingly: uploading blocks from the CPU to GPU, performing computations on the GPU, and offloading them back to the CPU **block by block**. This distinct offloading strategy necessitates entirely different memory management, scheduling, overlap, and AMP designs, tailored specifically to support the zeroth-order optimization approach.
>
> 3. BLOOM Models and More Datasets Evaluation.
>
> > BLOOM models are also open-source. Running only OPT models limits your showcase of the applicability of this model. Communication delays are not primary bottleneck for OPT, but may be so for other LLMs. Also, MeZO paper has shown results with multiple datasets; this paper shows results with only one dataset. This reviewer has carefully examined the supplementary material also.
>
> As noted in the manuscript, we chose the OPT model family for evaluation because it provides a wide range of model sizes (125M to 175B), allowing us to demonstrate the scalability of our approach. However, we acknowledge the value of including other open-source models such as BLOOM-176B and have conducted additional experiments on these models:
>
> **Table: More Experiment Results for BLOOM.** Instances of '-' in the table indicate scenarios where the corresponding method failed to execute due to memory constraints. The values in parentheses (x) represent the ratio of each measurement compared to the baseline MeZO (first column) configuration.
>
> | Model      |     |  GPU Memory Usage (MB) ↓ |    |     |     |  Throughput (tokens/sec) ↑ |    |     |
> |------------|-----------------------------|-----|-----|-----|-----------------------------|-----|-----|-----|
> |            | MeZO(32)                    | ZO-Offload(32) | MeZO(16) | ZO-Offload(16) | MeZO(32) | ZO-Offload(32) | MeZO(16) | ZO-Offload(16) |
> | BLOOM-176B | -                           | 49525 | -   | **24864** | -          | 14 | -   | **37** |
>
>
>
> In our original Appendix Section D, Table 4, we have documented the expanded scope of our experiments, which aligns with the methodologies described in the MeZO paper.
>
> **Table: Main results of ZO-Offloading precision on OPT-13B**
>
> | Method         | SST-2 (%) | RTE (%) | CB (%) | BoolQ (%) | WSC (%) | WIC (%) | MultiRC (%) |
> |----------------|-----------|---------|--------|-----------|---------|---------|-------------|
> | MeZO           | 91.4      | 66.1    | 67.9   | 67.6      | 63.5    | 61.1    | 60.1        |
> | ZO-Offloading  | 91.4      | 66.1    | 67.9   | 67.6      | 63.5    | 61.1    | 60.1        |

---

> ### Author Response · Authors · 2024-11-26
> **Response by Authors (Part2)**
>
> 4. On AMP and Communication Bottlenecks
>
> > Table 3: no benefit from using FP8, compared to FP16? Even FP16/BF16 are useful only for OPT-6.7B. This means that uploading/offloading low-precision data does not help much, which means the CPU-GPU transfer (Communication) is not a bottleneck. Actually, the real benefit of your technique will be clear when this communication is a bottleneck.
>
> Regarding low-precision data (FP8, FP16, BF16), we conducted additional experiments on larger models such as OPT-175B to further demonstrate the benefits of our method. These results show that the benefits of our approach become more pronounced as model size increases.
>
> **Table: Complete throughput (token/sec) results to validate AMP Mode.** AMP auto-cast with FP16 (top) and BF16 (below).
>
> | Model    | ZO-Offload (non-compress) | ZO-Offload (FP16) | ZO-Offload (BF16) | ZO-Offload (FP8) |
> |----------|---------------------------|-------------------|-------------------|------------------|
> | OPT-1.3B | 4827                      | 4770 (x0.988)     | 4760 (x0.986)     | 4802 (x0.995)    |
> | OPT-2.7B | 2811                      | 2974 (x1.058)     | 2974 (x1.058)     | 2997 (x1.066)    |
> | OPT-6.7B | 1271                      | 1641 (x1.291)     | 1641 (x1.291)     | 1662 (x1.308)    |
> | OPT-13B  | 561                       | 930 (x1.658)      | 930 (x1.658)      | 951 (x1.695)     |
> | OPT-30B  | 286                       | 416 (x1.455)      | 416 (x1.455)      | 425 (x1.486)     |
> | OPT-66B  | 127                       | 192 (x1.512)      | 192 (x1.512)      | 198 (x1.559)     |
> | OPT-175B | 43                        | 65 (x1.512)       | 65 (x1.512)       | 68 (x1.584)      |
> |----------|---------------------------|-------------------|-------------------|------------------|
> | OPT-1.3B | 4565                      | 4430 (x0.970)     | 4430 (x0.970)     | 4463 (x0.978)    |
> | OPT-2.7B | 2778                      | 2816 (x1.014)     | 2816 (x1.014)     | 2818 (x1.014)    |
> | OPT-6.7B | 1273                      | 1594 (x1.252)     | 1594 (x1.252)     | 1612 (x1.266)    |
> | OPT-13B  | 678                       | 910 (x1.342)      | 910 (x1.342)      | 924 (x1.363)     |
> | OPT-30B  | 285                       | 407 (x1.428)      | 407 (x1.428)      | 415 (x1.456)     |
> | OPT-66B  | 127                       | 188 (x1.480)      | 188 (x1.480)      | 194 (x1.528)     |
> | OPT-175B | 43                        | 64 (x1.488)       | 64 (x1.488)       | 67 (x1.565)      |
>
> 5. On CPU offloading
>
> > Experimentation is somewhat weak and the proposed ideas are not very novel. Many works have already been done on shuttling data between CPU and GPU memories.
>
> While CPU offloading is a well-established technique, the novelty of this work lies in the seamless integration of CPU offloading with zeroth-order optimization, enabling fine-tuning of extremely large models with significantly reduced resource requirements. We believe the focus of novelty should be on this combination rather than on CPU offloading as a standalone technique.

---

> ### Author Response · Authors · 2024-11-26
> **Response by Authors (Part3)**
>
> On Minor Weaknesses:
>
> > It may have been better to conduct ablation experiments on different CPU-GPU interconnect (PCIe) bandwidths/configurations, although the reviewer understands it is not easy. But some ablation could have been performed on GPU systems space.
>
> Thank you for understanding the challenges of conducting experiments across different hardware configurations. As you correctly noted, obtaining diverse cluster hardware settings is difficult in a university lab environment. We apologize for the limitation and have conducted experiments using the only available cluster hardware.
>
> > Figures 2 and 3 and 4 use white color for text, which gets faded (with color background) when printed.
> > Figure 1, instead of 0, X should have been shown. 0 is misleading. Figure 1 caption could mention that these values are observed for a single GPU, because with multiple GPUs, you can definitely find these numbers.
> > Figure 1 shows GPU memory or CPU memory?
>
> Thank you for pointing out the readability issue with the figures. We have revised Figures 1, 2, 3, and 4 to improve contrast and ensure clarity when printed. Additionally, Figure 1 shows GPU memory usage, and we have clarified this in the caption.
>
> > "previously unattainable with conventional methods" is redundant.
> > Incomplete phrase "Since CPU resources can be combined and offloaded to expand the memory and computational capacity of a single GPU"
>
> We have corrected the redundant phrase "previously unattainable with conventional methods" and completed the incomplete phrase to enhance the manuscript's clarity in the revised version.
>
> > Can you theoretically say: What is the largest OPT model that your technique can run on a 24GB GPU, assuming that any OPT size model is available?
>
> Our framework requires GPU memory for only three transformer blocks' parameters at any given time—one block for receiving parameters from the CPU, one block for computation, and one block for offloading back to the CPU. This characteristic enables scalability irrespective of the total number of blocks in the model.
>
> > Table 3 should have been shown all the way till OPT 175B.
>
> We have extended Table 3 to include results for OPT-175B as we showed above.
>
> > Is ZO same as ZO-SGD? You are using both the terms.
>
> Thank you for noting the inconsistency. In this paper, ZO-SGD and ZO refer to the same concept, and we have revised the manuscript for consistency.
>
> > Comparison could have been performed with other techniques, if possible.
>
> As the first work to combine CPU offloading with the zeroth-order (ZO) method, we believe there are no directly comparable prior works other than standard ZO methods. This unique combination forms the basis of our contributions.

---

> ### Author Response · Authors · 2024-11-26
> **Request for Futher Feedback and Score Raise**
>
> Dear reviewer, today is the last day to revise the manuscript. If your concerns have been resolved, I kindly ask you to consider **raising your score**. If not, feel free to share any remaining questions. Thank you!

---

### Official Review · Reviewer_ayNu · 2024-11-03

**Soundness:** 2
**Presentation:** 2
**Contribution:** 2
**Rating:** 3
**Confidence:** 5

**Summary:**

This paper presents a technique called ZO-Offloading which scales zeroth-order optimization methods of LLM finetuning by offloading model parameters to CPU. ZO-Offloading attempts to efficiently overlap the layer computation with transfers between CPU and GPU in order to minimize/avoid computation stall times. ZO-Offloading also integrates mixed-precision techniques and asynchronous model checkpointing to improve overall finetuning efficiency. The key result is the ability to finetune a 175B model using just 24GB of memory.

**Strengths:**

- The paper tackles an important AI democratization problem of reducing the hardware cost of using SOTA LLMs.
- Extending systems optimizations, such as offloading, to less-studied zeroth-order optimization techniques is empowering to the model scientists.

**Weaknesses:**

- The main weakness is that this work appears to overlook critical prior work such as ZeRO-Infinity. This oversight harms the paper in at least two major ways:
 1. There is no clear novelty in the parameter offloading approach since ZeRO-Infinity already demonstrated overlapping parameter offloading with forward (and backward) pass.
 2. The claim that finetuning 175B model using 24GB is unprecedented given that ZeRO-Infinity enables finetuning 1T model with Adam using 32GB.

[**Suggestion**]: To address the above concern, the authors should compare ZO-Offloading to ZeRO-Infinity, highlighting any key differences or improvements. Also, authors should revise their claims about novelty and unprecedented capabilities in light of ZeRO-Infinity's achievements.

- The asynchronous checkpointing appears to be missing from the evaluation section, making it difficult to appreciate the efficiency or effectiveness.

[**Suggestion**]: This concern can be addressed by updating the evaluation with results comparing asynchronous checkpointing and the baseline synchronous checkpointing. A useful evaluation metric to report would be training slowdown of checkpointing across different model sizes.

- Given that zeroth-order optimization requires only forward pass, I think comparison with the prior offloading inference work like FlexGen or ZeRO-Inference (another overlooked prior work) would be appropriate. Such comparisons could focus on forward pass efficiency.

[**Suggestion**]: To address this concern, the authors should include a comparative analysis table or graph that shows forward pass efficiency metrics (e.g., throughput, latency) for ZO-Offloading versus FlexGen and ZeRO-Inference across different model sizes (and perhaps batch sizes). Since this is a finetuning scenario, throughput comparison is probably most useful.

**Questions:**

1. Does the dual forward computation of block i occur before that of i+1? If so, how is block i updated based on loss information? (Figure 2).
2. What is the hardware environment of evaluation?

---

> ### Author Response · Authors · 2024-11-26
> **Response by Authors**
>
> Thank you for your thoughtful feedback and suggestions. We appreciate your insights and would like to address the concerns raised regarding prior work comparisons and novelty.
>
> 1. On ZeRO-Infinity.
>
> > The main weakness is that this work appears to overlook critical prior work such as ZeRO-Infinity. This oversight harms the paper in at least two major ways:
> > 1. There is no clear novelty in the parameter offloading approach since ZeRO-Infinity already demonstrated overlapping parameter offloading with forward (and backward) pass.
> > 2. The claim that finetuning 175B model using 24GB is unprecedented given that ZeRO-Infinity enables finetuning 1T model with Adam using 32GB.
>
> It is important to clarify that ZeRO-Infinity **primarily targets distributed multi-GPU fine-tuning scenarios and the foundational ZeRO algorithm is intrinsically designed for distributed environments**. ZeRO-Infinity extends this by incorporating CPU offloading for first-order optimizers, which includes gradient, activation value, and optimizer state offloading.
>
> However, our ZO-Offloading approach is fundamentally different and not compatible with ZeRO-Infinity's methods for several reasons: 1) Unlike ZeRO-Infinity, ZO-Offloading is designed for **single GPU scenarios**, which presents unique challenges and opportunities in terms of computational and memory resource management. 2) **ZeRO-Infinity's offloading optimizations are tailored for first-order optimizers and do not support the unique requirements of zeroth-order methods**. In zeroth-order optimization, there are no activation values or optimizer states to offload, and the gradient calculations are performed in-place, which is incompatible with ZeRO-Infinity's approach. 3) ZO-Offloading introduces a novel offloading strategy where transformer parameters are divided into blocks and processed in a streaming manner—uploading blocks from the CPU to the GPU, performing computations, and then offloading them back to the CPU block by block. This necessitates a completely different approach to memory management, scheduling, overlap, and AMP designs that are specifically tailored to support the zeroth-order optimization framework.
>
> Also you may misunderstand ZeRO-Infinity's memory requirements. The ZeRO-Infinity paper mentions using a **V100 DGX-2** node to finetune a 1T model. A V100 DGX-2 node consists of **16** V100 GPUs, each with 32GB of memory, for a total of **512GB** of GPU memory. **Translating this setup to a 175B model would still require approximately 100GB of GPU memory, which is far above the 24GB requirement in our framework**. Thus, our claim of unprecedented fine-tuning capabilities using only 24GB of GPU memory remains valid.
>
> 2. On Asynchronous Checkpointing
>
> > The asynchronous checkpointing appears to be missing from the evaluation section, making it difficult to appreciate the efficiency or effectiveness.
>
> **We have experimental results on the checkpointing**.
> This is an extension rather than a primary contribution of this work, so we leave its evaluation in the original Appendix Table 6.
>
> 3. On FlexGen and ZeRO-Inference
>
> > Given that zeroth-order optimization requires only forward pass, I think comparison with the prior offloading inference work like FlexGen or ZeRO-Inference (another overlooked prior work) would be appropriate. Such comparisons could focus on forward pass efficiency.
>
> We acknowledge the value of prior work like FlexGen and ZeRO-Inference; however, both are **inference** frameworks that focus on optimizing tasks such as prefill, decode, and KV-cache management. In contrast, ZO-Offloading is a **fine-tuning framework** that incorporates **dual-forward passes with parameter perturbations, gradient computation, and parameter updates**. These fundamental differences make a direct comparison with inference frameworks less relevant. Fine-tuning and inference workflows have distinct objectives and computational requirements, which is why we have not included such comparisons.
>
> ## For the questions:
>
> > Does the dual forward computation of block i occur before that of i+1? If so, how is block i updated based on loss information? (Figure 2).
>
> The parameter updates utilize the projected gradient from the j-1 th iteration to update the parameters in the j-th iteration. Therefore, even though the dual forward computation of block i occurs before that of block i+1, the parameters are updated accurately.
>
> > What is the hardware environment of evaluation?
>
> The experimental environments are detailed in Appendix Section C. We used NVIDIA A100 GPUs with 80GB of memory and AMD Milan CPUs.

---

> ### Author Response · Authors · 2024-11-26
> **Request for Futher Feedback and Score Raise**
>
> Dear reviewer, today is the last day to revise the manuscript. If your concerns have been resolved, I kindly ask you to consider **raising your score**. If not, feel free to share any remaining questions. Thank you!

---

### Official Review · Reviewer_qJqg · 2024-11-04

**Soundness:** 1
**Presentation:** 2
**Contribution:** 1
**Rating:** 8
**Confidence:** 4

**Summary:**

This paper proposes ZO-Offloading, a framework that efficiently fine-tunes large language models (LLMs) on GPUs and CPUs by leveraging zeroth-order (ZO) optimization. ZO-Offloading alleviates the memory requirements of standard first-order optimizers by using two forward passes instead of backpropagation, eliminating the need for activation caching. It optimizes resource usage by minimizing GPU idle times by offloading model parameters between the GPU and the auxiliary memory of CPU. It also enables low-bit data transfers and synchronous checkpointing to optimize against memory space.

**Strengths:**

1. Devise a dynamic workload scheduler to arrange overlaps of computation and communication, including mechanism such as reusable one block space and asynchronous checkpointing that have industrial value.
2. Offloading partial GPU memory to auxiliary CPU memory fit in models of large capacities.

**Weaknesses:**

As Table 1 conveys memory usages and throughputs of ZO-Offloading and other baselines, authors may consider showing overlapped computation and communication time, the ratio of overlaps compared with other ZO baselines. Doing so gives a clear picture of the advantages of ZO-Offloading's dynamic overlap scheduling.

**Questions:**

The authors are encouraged to elaborate preemptive parameter updates by 1). showing a diagram regarding this mechanism, especially how it "halves the usage of interconnection bandwidth" compared with traditional two data transfer streams.

---

> ### Author Response · Authors · 2024-11-26
> **Response by Authors**
>
> Thank you for your encouraging feedback and high score. We greatly appreciate your recognition of the contributions and industrial value of our work. Your comments provide valuable insights, and we are happy to address the points raised.
>
> ## On Weakness
>
> > As Table 1 conveys memory usages and throughputs of ZO-Offloading and other baselines, authors may consider showing overlapped computation and communication time, the ratio of overlaps compared with other ZO baselines. Doing so gives a clear picture of the advantages of ZO-Offloading's dynamic overlap scheduling.
>
> As we know, ZO-Offloading is the first method to combine CPU offloading with zeroth-order optimization. Traditional ZO baselines do not include CPU offloading communication time since they do not involve offloading operations. Thus, a direct comparison of overlap ratios with other ZO baselines is not applicable. That said, we are open to including a detailed breakdown of the overlap efficiency within ZO-Offloading to further illustrate the advantages of our dynamic overlap scheduling.
>
> ## On Question
>
> > The authors are encouraged to elaborate preemptive parameter updates by 1). showing a diagram regarding this mechanism, especially how it "halves the usage of interconnection bandwidth" compared with traditional two data transfer streams.
>
> Your suggestion to include a diagram for preemptive parameter updates is well-taken, and we incorporated this in Appendix Section B Figure 7 of the revised version. To elaborate briefly, parameter updates in traditional training typically occur after the gradient computation. Without preemptive updates, parameters would require separate upload and download operations during this step. By integrating parameter updates with preemptive upload operations, i.e. the parameters in the j th iteration will be updated by the projected gradient from the j-1 th iteration, our method eliminates one full round of data transfer, effectively halving the interconnection bandwidth usage for parameter updates. This mechanism plays a key role in optimizing communication efficiency in ZO-Offloading.

---

### Note · Authors · 2024-12-01

I have read and agree with the venue's withdrawal policy on behalf of myself and my co-authors.